# Hmgb2 improves astrocyte to neuron conversion by increasing the chromatin accessibility of genes associated with neuronal maturation in a proneuronal factor-dependent manner

Priya Maddhesiya[1,2,3†], Tjasa Lepko[1,2,3], Andrea Steiner-Mezzardi[3], Julia Schneider[1,4], Veronika Schwarz[1,2,3], Juliane Merl-Pham[5], Finja Berger[1,2,3], Stefanie M. Hauck[5], Lorenza Ronfani[6], Marco Bianchi[6,7], Tatiana Simon[8], Anthodesmi Krontira[3,8], Giacomo Masserdotti[3,8], Magdalena Götz[3,8,9] and Jovica Ninkovic[1,2,3,4,9*]

†Priya Maddhesiya and Tjasa Lepko contributed equally to this work.

*Correspondence:
jovica.ninkovic@helmholtz-munich.de

[1] Department of Cell Biology and Anatomy, Biomedical Center Munich (BMC), Medical Faculty, LMU, Munich, Germany
Full list of author information is available at the end of the article

## Abstract

**Background:** Direct conversion of reactive glial cells to neurons is a promising avenue for neuronal replacement therapies after brain injury or neurodegeneration. The overexpression of neurogenic fate determinants in glial cells results in conversion to neurons. For repair purposes, the conversion should ideally be induced in the pathology-induced neuroinflammatory environment. However, very little is known regarding the influence of the injury-induced neuroinflammatory environment and released growth factors on the direct conversion process.

**Results:** We establish a new in vitro culture system of postnatal astrocytes without epidermal growth factor that reflects the direct conversion rate in the injured, neuroinflammatory environment in vivo. We demonstrate that the growth factor combination corresponding to the injured environment defines the ability of glia to be directly converted to neurons. Using this culture system, we show that chromatin structural protein high mobility group box 2 (HMGB2) regulates the direct conversion rate downstream of the growth factor combination. We further demonstrate that Hmgb2 cooperates with neurogenic fate determinants, such as Neurog2, in opening chromatin at the loci of genes regulating neuronal maturation and synapse formation. Consequently, early chromatin rearrangements occur during direct fate conversion and are necessary for full fate conversion.

**Conclusions:** Our data demonstrate novel growth factor-controlled regulation of gene expression during direct fate conversion. This regulation is crucial for proper maturation of induced neurons and could be targeted to improve the repair process.

## Background

Innovative approaches to stimulate tissue regeneration and functional restoration of the central nervous system are required, because the adult mammalian brain has limited ability to replace lost neurons [1–4]. Direct conversion of glial cells to neurons (induced neurons, iN) is a promising avenue for successful repair [2, 5, 6]. The overexpression of several neurogenic factors, alone or in combination, induces the conversion of several cell types, including astrocytes, pericytes, oligodendrocyte progenitors, and fibroblasts, into post-mitotic neurons with different well-defined neurotransmitter identities [7–24]. These strong inducers of the neurogenic fate are transcription factors (TFs) that specify neuronal fate during development [7]. Many of these TFs have recently been shown to have pioneering factor activity and to bind closed chromatin configurations [5, 25, 26]. Indeed, recent insights regarding the fundamentals of neuronal fate specification have revealed that changes in chromatin structure might be a key factor in the stable acquisition of neuronal fate [27, 28], in line with the pioneering activity of fate determinants inducing fate conversion. Despite their remarkable strength, defined single pioneering TFs (e.g., Neurog2) cannot successfully reprogram some starting cell types or cell states induced by culturing conditions [14]. The inability of Neurog2 to activate gene expression has been associated with epigenetic silencing of target loci [14, 29]. Interestingly, forskolin (an agonist of adenylyl cyclase) and dorsomorphin (an inhibitor of BMP signaling) enhance the chromatin accessibility mediated by Neurog2, thus suggesting that additional pathways contribute to Neurog2's trailblazing properties [30, 31]. In fact, treating Neurog2-expressing cells with these small molecules results in chromatin opening at a substantial number of sites, including CRE half-sites or HMG box motifs [30]. Thus, small molecules or a combination of other TFs may be necessary to induce successful or efficient reprogramming, depending on the starting populations, although Neurog2 is a pioneer factor that can overcome the lineage barrier. In addition to several factors associated with chromatin, microRNAs and small molecules have been found to improve the conversion efficiency and maturation status of reprogrammed neurons despite being unable to induce conversion on their own [12, 15, 32, 33]. These findings support a model in which multi-level lineage barriers maintain cell identity and must be overcome for cells to acquire neuronal fate adequate for repair purposes. Comprehensive understanding of these barriers is at the core of successful iN generation and the functional restoration of the damaged CNS.

Importantly, most of these barriers have been identified through the use of defined and stable in vitro systems. However, for repair purposes, iNs must be generated in the injured environment. The intricacy of the injured milieu is an obstacle to understanding the molecular mechanisms of direct neuronal conversion in vivo. Injury triggers the release of several signaling factors with precise temporal resolution that can either resolve or strengthen the lineage barriers [34]. For example, epidermal growth factor (EGF) levels spike within 24 h after brain injury and remain elevated for 3 days before returning to baseline. In contrast, basic fibroblast growth factor (bFGF) levels begin to rise 4 h after damage and remain elevated for 14 days [34]. Infusion of bFGF into the brain after traumatic brain injury, for example, greatly enhances cognitive performance in animals by increasing neurogenesis [35]. Additionally, EGF infusion enhances neurogenesis via enlargement of the neurogenic precursor pool in the neurogenic niche after

ischemia injury [36]. Moreover, forced Neurog2 expression in glial cells, along with the bFGF2 and EGF growth factors, enhances neuronal reprogramming in vivo [10]. Importantly, EGF receptor (EGFR) signaling has been proposed to regulate both global chromatin state and the accessibility of specific loci [37]. Furthermore, interaction of EGFR signaling and chromatin remodelers from the SWI/SNF family is critical for the expansion of beta cells after pancreas injury [38]. Similarly, FGF signaling orchestrates chromatin organization during neuronal differentiation [39]. Together, environmental signals are likely to be integrated into the lineage barriers defining the propensity of starting glial cells to be converted to postmitotic neurons.

To investigate the embedding of growth factors in lineage barriers relevant to in vivo direct neuronal reprogramming after brain injury, we developed an in vitro model with altered growth factor composition. We showed that, in this model, neurogenic fate determinants induced astrocyte to neuron conversion with a diminished efficiency comparable to the conversion rate observed in vivo. This system allowed us to identify Hmgb2 as a novel regulator in the context of direct astrocyte to neuron conversion. We showed that high levels of Hmgb2 alleviate the lineage barrier and promote efficient establishment of neuronal fate. Our data suggest that Hmgb2-dependent chromatin opening of regulatory elements controls the expression of neuronal maturation genes and enables the establishment of the full neurogenic program, thereby resulting in efficient astrocyte to neuron conversion.

## Results

### Growth factors shape the lineage barriers to glia to neuron conversion

After brain injury, levels of EGF peak within the first 24 h and return to baseline levels 3 days post injury (dpi). In contrast, FGF levels increase by 4 h after injury and persist until 14 dpi [34]. To mimic the dynamics in the in vivo environment, we cultured astrocytes, obtained from postnatal murine cerebral cortex (P5–P7) for 10 days in the presence of only bFGF, then compared the direct conversion rates to neurons in this culture with the conversion efficiency in the widely used culture conditions containing both EGF and bFGF [40, 41]. To convert astrocytes into neurons, we transduced cells with an MLV-based retrovirus for expression of the neurogenic TFs reported to reprogram astrocytes (Neurog2, Pou3f2 or Sox11; Fig. 1a) in vitro and a fluorescent reporter protein. The expression of the fluorescent reporter protein was used to identify the transduced cells. The identity of the transduced cells was probed 7 days after viral transduction (days in vitro (div); Fig. 1a). Only cells expressing doublecortin (DCX) and having at least one process longer than three cell somata diameters were identified as neuronal cells, according to Gascon et al. [42] (Fig. 1b, c). The transduction of astrocytes with control viruses for expression of either GFP or dsRed did not induce glia to neuron conversion in any culturing conditions (Additional file 1: Fig. S1a-d). In contrast, the transduction of astrocytes isolated from EGF + bFGF culture with several neurogenic fate determinants did induce their conversion, and neurons at different maturation stages (on the basis of the complexity of their processes) were observed after 7 div (Fig. 1b, d). Interestingly, neither Neurog2 nor Pou3f2 induced the direct conversion of astrocytes grown in the presence of only bFGF, whereas the culturing conditions did not significantly alter the conversion by overexpression of Sox 11 (Fig. 1d). Because the culture condition with

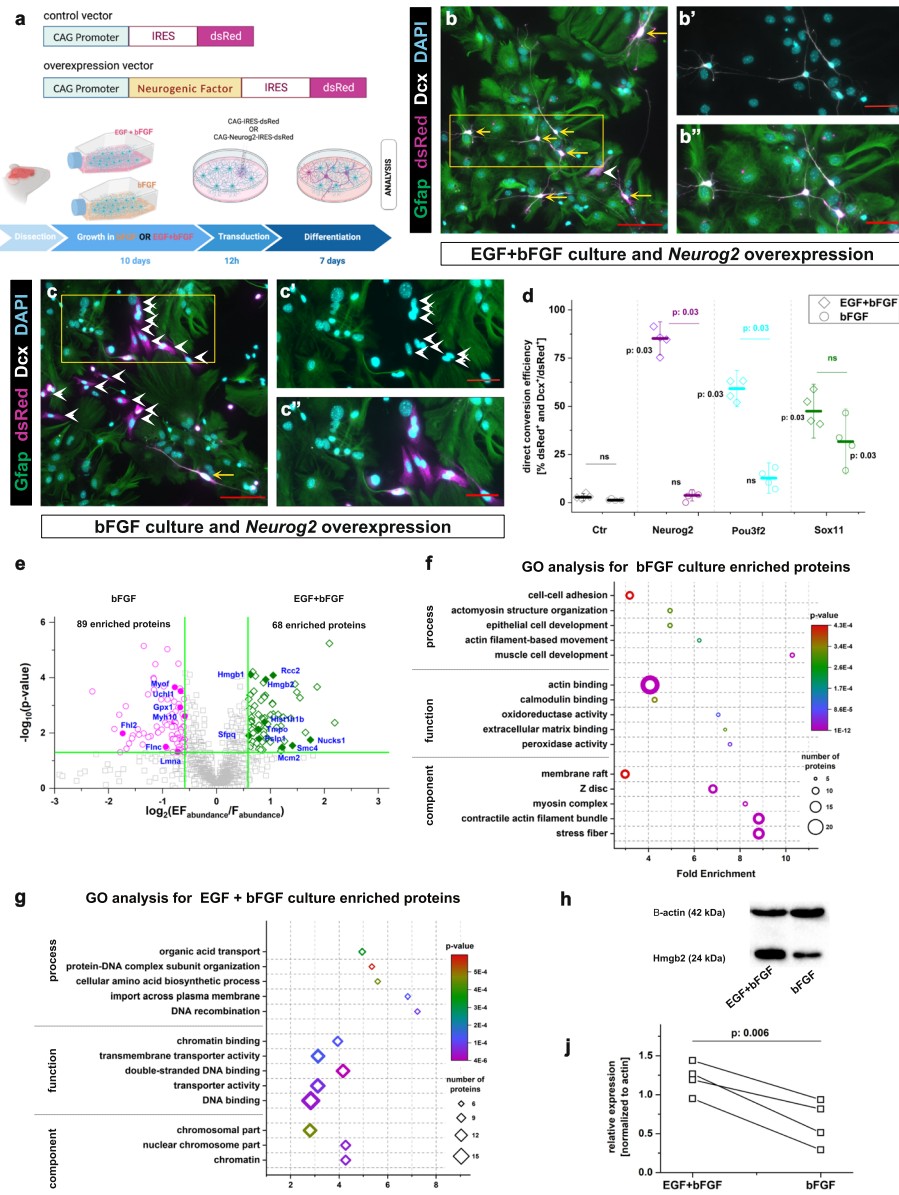

**Fig. 1** Astrocyte growth conditions define the rate of direct astrocyte to neuron conversion **a** Schemes depicting viral vector design and the experimental paradigm used for astrocyte to neuron conversion. **b,c"** Micrographs illustrating the identity of Neurog2-transduced cells 7 days after transduction in the EGF + bFGF (**b**) and bFGF (**c**) culture conditions. **b', b", c',** and **c"** are magnifications of boxed areas in **b** and **c**, respectively. Yellow arrows indicate successfully converted cells, whereas white arrowheads indicate cells failing to convert. Scale bars: 100 μm in **b** and **c**; 50 μm in **b', b", c',** and **c". d** Dot plot depicting the proportion of transduced cells converting to neurons in EGF + bFGF and bFGF cultures 7 days after transduction with different neurogenic fate determinants. Data are shown as median ± IQR; each single dot represents an independent biological replicate. Significance was tested with two-tailed Mann-Whitney test. *p*-values: black font corresponds to the comparison to the control and colored to the comparison between EGF + bFGF and bFGF. **e** Volcano plot depicting proteins enriched in astrocytes cultured in bFGF (magenta circles) and EGF + bFGF (green diamonds) culture conditions (fold change > 1.5; *p* value < 0.05). **f, g** Plots depicting the top five enriched GO terms in protein sets enriched in bFGF (**f**) and EGF + bFGF (**g**) cultures. **h** Western blot depicting levels of Hmgb2 protein in EGF + bFGF and bFGF astrocyte cultures. **j** Dot plot showing the relative levels of Hmgb2 (normalized to actin) in EGF + bFGF and bFGF cultures. Data are shown as median ± IQR; single dots represent independent biological replicates. Paired *t*-test was used for the significance test. Abbreviation: GO, Gene Ontology

bFGF contained only half the usual growth factors, we assessed the conversion rate of cultures containing only EGF. Importantly, Neurog2 induced the conversion of astrocytes grown with only EGF at the same rate as astrocytes grown in EGF + bFGF culture medium (Additional file1: Fig. S1d-f), in line with the specific role of EGF in decreasing the lineage barrier and promoting direct neuronal conversion.

This difference in direct conversion could be explained by the selection of particular cell types during astrocyte expansion with growth factors. Therefore, we assessed the identity of the transduced cells 24 h after transduction by using immunocytochemistry (Additional file 1: Fig. S2a). Most cells expressed the astrocyte marker S100β in both culture conditions, without any significant differences (Additional file 1: Fig. S2b, c, f). Similarly, we did not observe any differences in the proportion of GFAP + cells (Additional file 1: Fig. S2d-f). In line with reports that astrocytes in vitro express the TF Olig2 [43], most cells in both culture conditions expressed Olig2 (Additional file 1: Fig. S2g-i). Moreover, we observed only a small proportion of DCX + neuronal progenitors or αSMA + pericytes in both cultures (Additional file 1: Fig. S2d-i). The mRNA levels of neuronal progenitor genes, such as Meis2, Pax6, and Dcx, showed no significant differences between the two conditions and were almost undetectable compared to their levels in the neurogenic E14 cerebral cortex (Additional file 1: Fig. S3a, b). These results indicate that the cellular compositions of the cultures are comparable based on the analyzed marker expression. Interestingly, we observed lower proliferation rates of astrocytes grown in bFGF than EGF + bFGF conditions, on the basis of the expression of Ki67 or pH3 (Additional file 1: Fig. S2j-n). This finding suggested that bFGF-grown astrocytes might further differentiate, epigenetically silence neuronal loci and become less prone to direct conversion, as previously shown for long-term astrocyte cultures [44]. To examine this possibility, we cultured astrocytes for 7 days in bFGF culture conditions, added EGF and grew astrocytes for an additional 7 days with EGF + bFGF (Additional file 1: Fig. S4a). The conversion rate of these astrocytes was compared with that of astrocytes cultured in either EGF + bFGF or bFGF for 14 days (Additional file 1: Fig. S4b, f). As expected, longer culturing of cells in either bFGF or EGF + bFGF decreased the direct reprogramming rate (Additional file 1: Fig. S4f), as previously described [44]. However, the post-culturing of initially bFGF-grown astrocytes in EGF + bFGF for 7 days improved their reprogrammability, and we observed no differences in the proportions of generated neurons compared with astrocytes continuously cultured in EGF + bFGF (Additional file 1: Fig. S4b, c, f). Moreover, the conversion rate of EGF + bFGF-grown astrocytes decreased after culturing in bFGF for 7 days, and no differences were observed between this culture and continuously bFGF cultured astrocytes (Additional file 1: Fig. S4d-f).

### High mobility group box 2 (Hmgb2) levels are decreased in bFGF astrocyte culture

To identify factors responsible for maintaining the astrocytic lineage barrier, we performed label-free LC–MS/MS-based proteome analysis of astrocytes cultured with either bFGF or EGF + bFGF for 10 days. In total, we detected approximately 1700 proteins, of which 157 showed differences in levels between culture conditions (1.5-fold change, $p < 0.05$): 68 significantly enriched in the EGF + bFGF culture and 89 significantly enriched in the bFGF culture (Fig. 1e). Gene Ontology (GO) analysis revealed an enrichment of cytoskeleton-associated processes in the protein set enriched in the

bFGF-grown culture (Fig. 1f; Additional File 2: Table S1a), whereas transport across the mitochondrial membrane, metabolic processes, and chromatin-associated processes were enriched in the EGF + bFGF induced proteome (twofold enrichment, $p < 0.05$; Fig. 1g, Additional File 2: Table S1b). These data are in line with recent evidence indicating that changes in the mitochondrial proteome during astroglia to neuron conversion determine the extent of the direct conversion [31]. Moreover, because chromatin state has been reported to regulate lineage barriers in reprogramming [42, 45–49], we searched for chromatin-associated factors differentially enriched between culture systems. The chromatin architectural protein Hmgb2 was 1.88-fold enriched in EGF + bFGF compared with bFGF cultures (Fig. 1e). This enrichment was confirmed by western blotting (Fig. 1h, j) and analysis of the HMGB2 signal at the single cell level (Additional file 1: Fig. S3c-g). Notably, the reprogrammability of the astrocyte culture in growth factor swapping experiments correlated with HMGB2 levels. Specifically, transitioning astrocytes from bFGF to EGF + bFGF increased HMGB2 levels, while switching from EGF + bFGF to bFGF decreased them (Additional file 1: Fig. S5). These findings, along with the cell identity marker analysis and culture composition experiments, indicate that growth factor conditions determine astrocytic lineage barriers and, consequently, the efficiency of direct conversion to neurons through neurogenic factor overexpression.

In the adult mouse brain, Hmgb2 is specifically expressed in cells committed to the neurogenic lineage (transit amplifying progenitors, neuroblasts) in both neurogenic niches [50]. Traumatic brain injury induces HMGB2 expression in a subset of reactive astrocytes (Additional file 1: Fig. S6). HMGB2 expression was detected as early as 1 day post-injury (dpi), peaked at 3–5 dpi, and declined by 7 dpi (Additional file 1: Fig. S6d-g), mirroring the pattern of EGF expression after brain injury [34]. Additionally, applying EGF to the uninjured adult brain induced HMGB2 expression within 24 h (Additional file 1: Fig. S6h-j). These findings suggest that EGF-induced HMGB2 may play a key role in enhancing direct conversion in the EGF + bFGF culture.

### Hmgb2 levels define the rate of direct astrocyte to neuron conversion

To test whether Hmgb2 might have functional relevance in fate conversion, we transduced astrocytes, grown for 10 days in medium containing either EGF + bFGF or bFGF, with Hmgb2-encoding retrovirus (Fig. 2a), and assessed the identity of the transduced cells 7 days later, on the basis of DCX expression and cell morphology. Overexpression of Hmgb2 did not alter cell identity in either culture condition (Fig. 2b–e). Most cells retained their astrocyte identity and expressed GFAP (Fig. 2e). However, when we co-transduced the bFGF-grown astrocytes with retroviruses for expression of Neurog2-dsRED and Hmgb2-GFP, we observed a 2.5-fold greater conversion rate in the co-transduced cells than cells transduced with Neurog2 only (Fig. 2c, d). Interestingly, the co-overexpression of Neurog2 + Hmgb2 did not further improve the conversion of EGF + bFGF-grown astrocytes, because the conversion rate of Neurog2 + Hmgb2 co-transduced astrocytes was comparable to that of Neurog2-transduced astrocytes in this culture condition (Fig. 2b, d).

Improvement in the Neurog2-mediated conversion rate of bFGF-grown astrocytes prompted us to investigate whether this improvement might be factor-specific. Therefore, we assessed the effect of Hmgb2 overexpression on Pou3f2-mediated fate

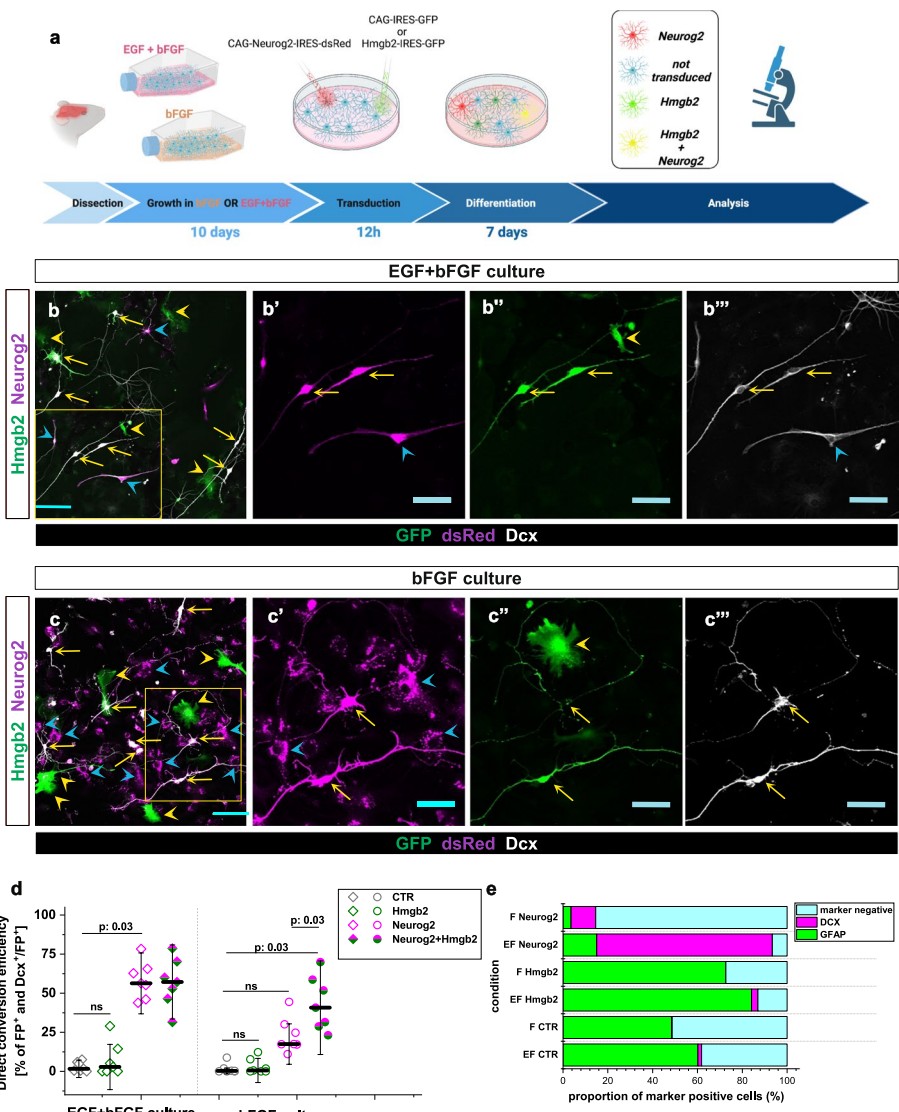

**Fig. 2** Hmgb2 is sufficient for successful Neurog2-mediated direct astrocyte to neuron conversion **a** Scheme depicting the experimental paradigm used for astrocyte to neuron conversion. **b,c'''**) Micrographs showing the identity of Neurog2- and Hmgb2-expressing virally transduced cells 7 days after transduction in EGF + bFGF (**a**) and bFGF cultures (**b**). **b'**, **b''**, **b'''**, **c'**, **c''**, and **c'''** are magnifications of the boxed areas in **a** and **b**, respectively. Yellow arrows indicate co-transduced cells expressing Neurog2 and Hmgb2, yellow arrowheads indicate cells transduced only with Hmgb2-encoding virus, and blue arrowheads indicate cells transduced with only Neurog2-encoding virus. Scale bars: 100 μm in **b** and **c**; 50 μm in **b'**. **b''**, **b'''**, **c'**, **c''**, and **cb'''**. **d** Dot plot depicting the proportion of transduced cells converting to neurons in EGF + bFGF and bFGF cultures 7 days after transduction. Data are shown as median ± IQR; single dots represent independent biological replicates. Significance was tested with two-tailed Mann-Whitney test. **e** Histogram depicting the identities of cells transduced with the indicated factors 7 days after transduction. Abbreviation: FP, fluorescent protein

conversion, given that the neurogenic capability of Pou3f2 was also diminished in bFGF-grown astrocytes (Fig. 1d). Similarly to the Neurog2-mediated conversion, the simultaneous overexpression of Hmgb2 and Pou3f2 in EGF + bFGF-grown astrocytes did not result in higher conversion rates, whereas the factor combination significantly

increased the conversion rate in bFGF-grown astrocytes (Additional file 1: Fig. S1g). Together, these data suggested that Hmgb2 does not induce direct conversion on its own but increases the ability of neurogenic factors to overcome the lineage barriers.

To determine whether HMGB2 is necessary for direct astrocyte-to-neuron conversion, we isolated astroglia from HMGB2-deficient mice (Hmgb2$^{MUT/MUT}$) and their siblings (Hmgb2$^{WT/MUT}$ and Hmgb2$^{WT/WT}$). These cells were cultured under direct conversion-permissive conditions (EGF + bFGF) (Fig. 3a). Consistent with earlier results (Fig. 1d), Neurog2 overexpression successfully induced direct conversion of Hmgb2$^{WT/WT}$ and Hmgb2$^{WT/MUT}$ astrocytes into neurons (Fig. 3b–d). However, the conversion rate was significantly lower in Hmgb2$^{MUT/MUT}$ astrocytes compared to their WT siblings (Fig. 3c, d). Despite this reduction, some conversion still occurred, suggesting the possibility that other HMGB family proteins might partially compensate for the absence of HMGB2. Since the HMGB2 protein family member HMGB1 was enriched in the EGF + bFGF culture, albeit at a lower level (Fig. 1e), we assessed HMGB1 expression in HMGB2-deficient astrocytes. Our analysis revealed no upregulation of HMGB1 in these cells (Fig. 3e–i), ruling out the possibility that increased HMGB1 levels compensate for the lack of HMGB2.

Additionally, neurons generated from HMGB2-deficient astrocytes displayed reduced neurite complexity compared to those derived from WT astrocytes (Fig. 3j, k). These findings support our hypothesis that HMGB2 levels are critical for maintaining the astrocytic lineage barrier, and increasing HMGB2 levels promotes direct astrocyte-to-neuron conversion.

### Prospero homeobox protein 1 (Prox1) overexpression improves direct glia to neuron conversion in FGF only culture

To understand the Hmgb2-dependent lineage barrier in direct glia to neuron conversion, we compared the transcriptional changes induced by Neurog2 overexpression in the bFGF and EGF + bFGF cultured cells 48 h after transduction. Cells transduced with different viruses were purified by FACS (Additional file 1: Fig. S7), and genes regulated by Neurog2 overexpression were compared. We identified differences in the expression of 443 genes (321 upregulated and 122 downregulated genes, fold change > 2, padj < 0.05) induced by Neurog2, as compared with that in control CAG-GFP virally transduced cells in the EGF + bFGF culture condition (Additional file 1: Fig. S8a, Additional File 3: Table S2). In the bFGF culture, Neurog2, as compared with the respective CAG-GFP transduced control, induced 171 genes (137 upregulated and 34 downregulated genes, fold change > 2, padj < 0.05) (Additional file 1: Fig. S8b, Additional File 3: Table S2). GO analysis (biological processes, fold enrichment > 2, and $p < 0.05$) of genes (321) upregulated in EGF + bFGF culture revealed enrichment in the terms nervous system development, neuronal differentiation, and migration (Fig. 4a; Additional File 4: Table S3a), in line with the ability of Neurog2 to successfully convert astroglia to neurons. Unexpectedly, the significantly enriched biological processes in the set of the 137 upregulated genes in the bFGF culture were also associated with regulation of neurogenesis, nervous system development, and synaptic signaling (Fig. 4b, Additional File 4: Table S3b), thereby indicating that Neurog2 overexpression at least partially induced the neuronal fate in astrocytes grown in the bFGF condition. Indeed, we observed that 96 genes were

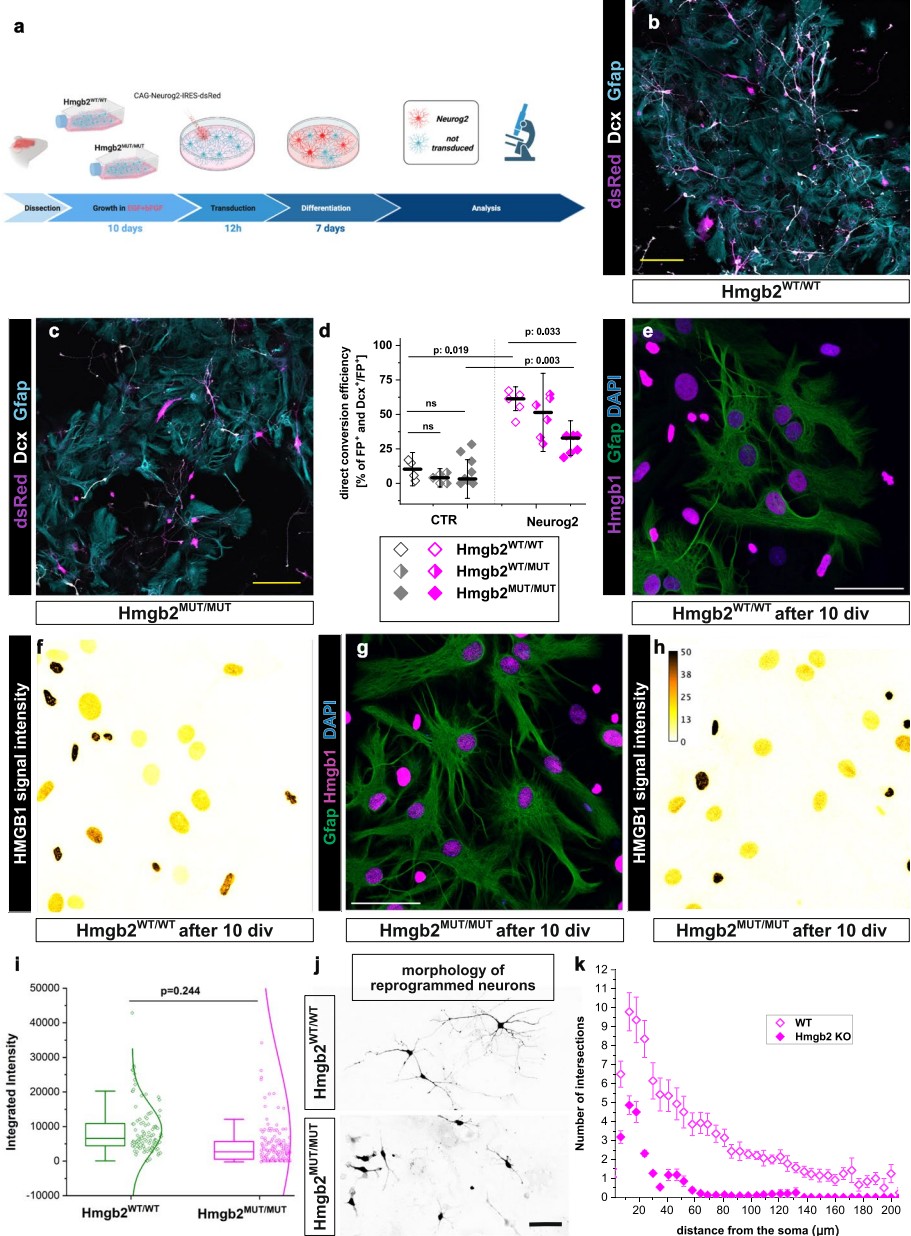

**Fig. 3** Hmgb2 is necessary for successful Neurog2-mediated direct astrocyte to neuron conversion **a** Scheme depicting the experimental paradigm used for astrocyte to neuron conversion. **b, c** Micrographs showing the identities of Neurog2-expressing virally transduced cells 7 days after transduction in EGF + bFGF culture of astrocytes derived from Hmgb2-deficient animals (**c**) and their siblings (**b**). Scale bars: 100 μm. (**d**) Dot plot depicting the proportion of Hmgb2-deficient or control cells converting to neurons 7 days after transduction with Neurog2. Data are shown as median ± IQR; single dots represent independent biological replicates. Significance was tested with two-tailed Mann-Whitney test. **e–h** Micrographs illustrating the expression of Hmgb1 in EGF + bFGF grown astrocytes from Hmgb2 deficient (**g**, **h**) animals and their siblings (**e**, **f**). **f**, **h** are pseudo colored images depicting expression level of Hmgb1 in single cells. Scale bar in **b–f**: 50 μm. **i** Dot plot showing the expression level of Hmgb1 in single cells. Single dots represent single cells from 4 independent biological replicates. Significance was tested with two-tailed Mann-Whitney test. **j** Representative images of ImageJ traces used for Sholl analysis of morphology of neurons reprogrammed from Hmgb2 deficient and WT astrocytes. Scale bar 50 μm. **k** Sholl analysis of induced neurons by overexpression of Neurog2 in Hmgb2-deficient and WT astrocyte (EGF + bFGF) culture 7 days after viral transduction

induced by Neurog2 in both bFGF and EGF + bFGF cultures (Fig. 4c) and were enriched in GO biological processes associated with regulation of neurogenesis, nervous system development, neuronal differentiation, and migration (Additional file 1: Fig. S8c; Additional File 4: Table S3c). In addition, in the bFGF culture, the 41 genes uniquely induced by Neurog2 (Fig. 4c) were associated with GO biological processes of cardiac muscle tissue development, leukocyte differentiation, response to lithium-ion and neurotransmitter receptor to the plasma membrane (Additional file 1: Fig. S8d; Additional File 4: Table S3d). These findings suggested that, in contrast to the EGF + bFGF culture, in the bFGF culture, Neurog2 induced other fates along with neuronal processes possibly interfering with the establishment of the neuronal identity [51]. Furthermore, we identified 225 uniquely Neurog2-induced genes in the EGF + bFGF culture (Fig. 4c) associated with the GO biological processes regulation of membrane potential and ephrin receptor pathway (Additional file 1: Fig. S8d), which regulate neuronal maturation and axonogenesis [52, 53]. Moreover, previously reported Neurog2-induced genes necessary for successful conversion, such as *Neurod4*, *Insm1*, *Hes6*, *Slit1*, *Sox11*, and *Gang4* [44] were upregulated in both cultures (Fig. 4d). Nevertheless, genes such as *Dscaml1*, *Prox1*, *Lrp8*, and *Shf* were induced in only the EGF + bFGF culture. Importantly, the co-expression of Neurog2 and Hmgb2 in bFGF-grown astrocytes induced the expression of these genes to levels similar to those detected in the Neurog2-transduced EGF + bFGF culture (Fig. 4d). Therefore, EGF-induced HMGB2 likely lowers the conversion barrier by regulating the expression of a small, specific set of genes that are crucial for the conversion process. To test this hypothesis, we selected one candidate gene, Prox1, and investigated whether its expression could help overcome the lineage barrier in the bFGF-only medium, which is characterized by low HMGB2 levels. We overexpressed Prox1 in the bFGF-cultured cells and observed only a small increase in the conversion rate (Fig. 4e). However, after the co-expression of Neurog2 and Prox1 in bFGF-cultured astrocytes, we observed a significant increase in the proportion of generated neurons similar to the conversion rate induced by Neurog2 in the EGF + bFGF culture and the bFGF-cultured astrocytes co-transduced

(See figure on next page.)

**Fig. 4** Neurog2 induces incomplete neuronal fate in bFGF culture **a**, **b** Plots depicting enriched GO biological process terms in gene sets induced by Neurog2 in EGF + bFGF culture (**a**) and bFGF culture (**b**) 48 h after viral transduction. Orange text represents the GO terms not associated with neuronal fate. Green and magenta text represent GO terms specifically enriched in EGF + bFGF culture and bFGF culture, respectively. **c** Venn diagram illustrating the overlap of Neurog2-induced transcripts in EGF + bFGF and bFGF culture 48 h after viral transduction. **d** Heat map showing Neurog2 − or Neurog2 + HMGB2-mediated induction of core neurogenic factors (according to Masserdotti et al., 2013) in EGF + bFGF and bFGF cultures. **e** Dot plot depicting the proportion of transduced cells converting to neurons in EGF + bFGF and bFGF cultures 7 days after transduction in Prox1 deficient or Prox-1 overexpressing cells. Data are shown as median ± IQR; single dots represent independent biological replicates. Significance was tested with two-tailed Mann-Whitney test. **f** Plot showing GO terms enriched in the gene set upregulated in bFGF culture by Neurog2 and Hmgb2 expression 48 h after viral transduction. GO terms in green text are also induced by Neurog2 alone in EGF + bFGF culture (panel **a**). Orange text represents the GO terms not associated with neuronal fate. **g** Venn diagram illustrating the overlap of Neurog2-induced transcripts in EGF + bFGF and bFGF culture with Neurog2 and Hmgb2-induced transcripts after overexpression in bFGF culture 48 h after viral transduction. **h** Plot depicting enriched GO biological process terms in gene sets induced in the reprogramming prone condition (46 genes set; Fig. 4g). GO terms in green text are also induced by Neurog2 alone in EGF + bFGF culture. Orange text represents the GO terms not associated with neuronal fate. Abbreviations: FP, fluorescent protein; GO, Gene Ontology

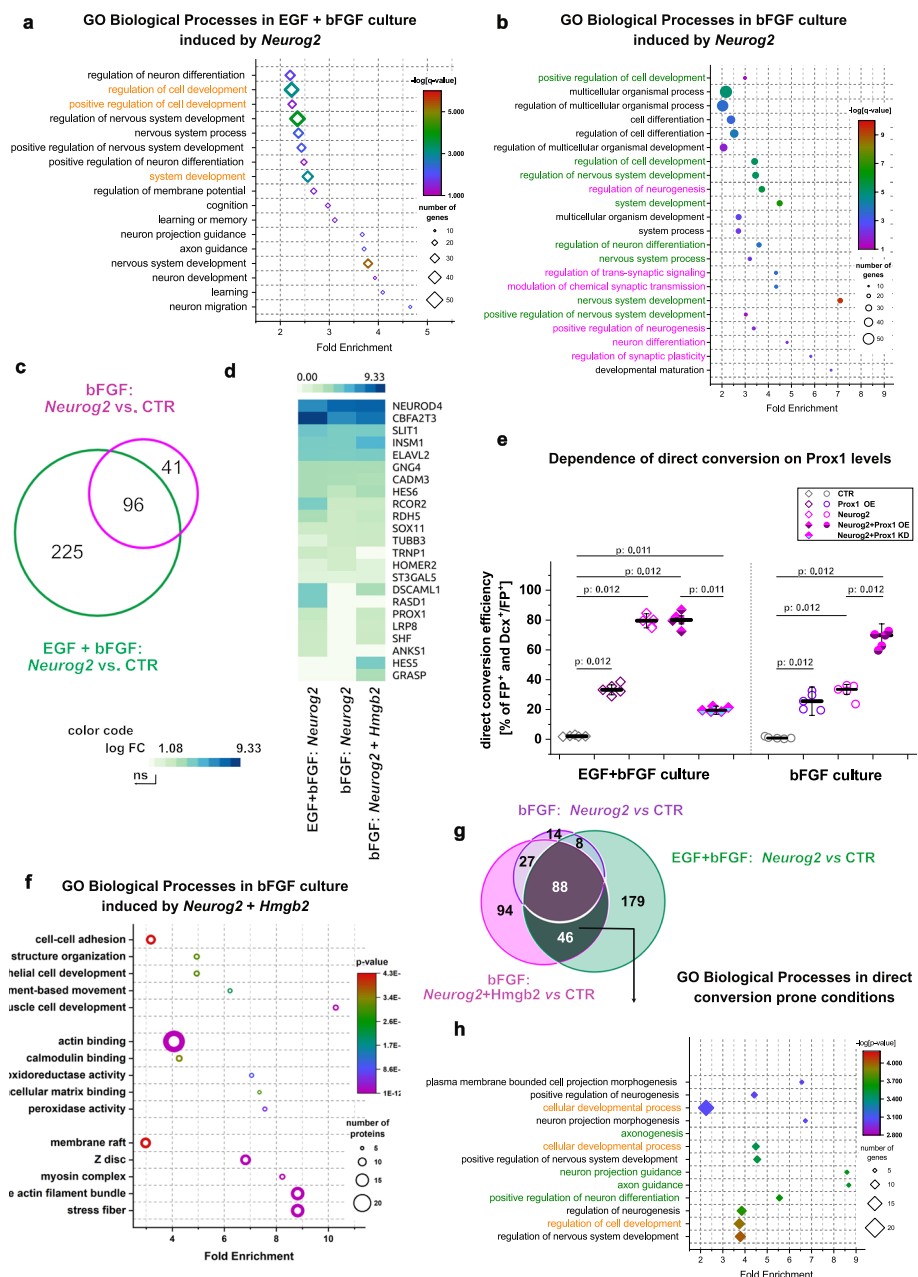

**Fig. 4** (See legend on previous page.)

with Neurog2 and Hmgb2 (Fig. 4e). Moreover, microRNA-mediated knockdown of Prox1 decreased the Neurog2-mediated conversion of EGF + bFGF cultured astrocytes (Fig. 4e), in line with previous reports [44]. This conversion rate was also comparable to the rate of Neurog2-mediated conversion of bFGF-cultured astrocytes (Fig. 4e).

### Hmgb2-dependent expression of a specific set of neuronal maturation genes is necessary for efficient direct glia to neuron conversion

Our data suggested that low Hmgb2 expression levels in the bFGF culture could decrease astrocyte to neuron conversion via several non-mutually

exclusive mechanisms: (a) failure to activate the full neurogenic program induced in EGF + bFGF culture, (b) prevention of the silencing of the conflicting alternative lineages, and (c) induction of a different neurogenic program from that in the EGF + bFGF culture. To directly test these possibilities, we analyzed the transcriptomic changes induced by the overexpression of Hmgb2 alone or in combination with Neurog2 in both bFGF and EGF + bFGF cultures.

Interestingly, Hmgb2 overexpression induced only several differentially expressed genes (DEGs) in either EGF + bFGF or bFGF cultures with respect to CAG-GFP control viral transduction ((Additional file 1: Fig. S8e, f; FC > 2, padj < 0.05), Additional File 3: Table S2). This transcriptomic analysis, together with the lack of change in the conversion rate after Hmgb2 overexpression in both bFGF and EGF + bFGF astrocytes (Fig. 2d), suggested that Hmgb2 did not implement any specific neurogenic program on its own. Notably, the overexpression of Hmgb2 together with Neurog2 in the bFGF culture, as compared with control viral transduction, induced 255 genes (Fig. 4g). This gene set was significantly enriched in GO biological processes associated with neural development, neuronal migration, axon guidance, and synaptic signaling (Fig. 4f; Additional File 4: Table S3e), similarly to the GO biological processes induced by Neurog2 alone in the EGF + bFGF condition (Fig. 4a). In addition, we observed downregulation of 164 genes (Additional File 3: Table S2.) enriched in regulation of cell adhesion, actin filament organization, stress fiber assembly, and negative regulation of protein phosphorylation (Additional file 1: Fig. S8g; Additional File 4: Table S3f). Since Neurog2 phosphorylation is linked to its ability to regulate gene expression [54–57], it is possible that HMGB2 promotes post-translational modifications, such as phosphorylation, which could enhance the efficiency of direct neuronal conversion. However, the downregulated genes were not associated with specific glial or alternative fates induced by Neurog2 in the bFGF culture (Additional file 1: Fig. S8g).

To determine whether the dual overexpression of Neurog2 + Hmgb2 might trigger similar transcriptional programs in the bFGF culture and the Neurog2-transduced the EGF + bFGF culture, we compared induced genes among three conditions: reprogramming prone culture (EGF + bFGF transduced with Neurog2 vs control virus), reprogramming resistant culture (bFGF transduced with Neurog2 vs control virus), and revived reprogramming culture (bFGF transduced with Neurog2 + Hmgb2 vs control virus). We identified 88 genes that were shared across all three conditions (Fig. 4g) and were enriched in GO biological processes associated with neurogenesis, neuronal differentiation and migration, and trans-synaptic signaling (Additional file 1: Fig. S8h; Additional file 4:Table S3g), in line with our findings that all conditions at least partially induced the neurogenic program. Furthermore, 46 genes (for example, *Prox1*, *Lrp8*, *Shf*, and *Dscaml1*) were shared exclusively between the reprogramming prone conditions (bFGF Neurog2 + Hmgb2 and EGF + bFGF Neurog2). This gene set was enriched in GO biological processes associated with axonogenesis, positive regulation of neurogenesis, neuron projection guidance, and nervous system development (Fig. 4h; Additional file 4: Table S3h), thus implying that the upregulation of genes induced by the simultaneous overexpression of Neurog2 and Hmgb2 in the bFGF culture are associated with the acquisition of a more mature neuronal phenotype.

Together, our data suggested that the Hmgb2 protein aids in implementing the Neurog2-dependent, neurogenic program in astrocytes by facilitating the induction of a specific set of neurogenic, neuronal maturation-associated genes.

### Hmgb2 increases the chromatin accessibility of regions associated with the neurogenic program

We hypothesized that the establishment of the full neurogenic program by high levels of Hmgb2 is associated with Hmgb2-dependent chromatin changes. Therefore, we performed assay for transposase-accessible chromatin with high-throughput sequencing (ATAC-seq) on the cells from the same sorting samples used to generate transcriptomic libraries (Additional file 1: Fig. S7). We first examined the genome-wide chromatin accessibility profile at transcription start sites (TSSs ± 3.0 kb) in both bFGF and EGF + bFGF cultures after the overexpression of Hmgb2, Neurog2, Neurog2 + Hmgb2, and CAG-GFP control. The accessibility profile of Hmgb2 overexpressing astrocytes was comparable to that of the control regardless of the culture condition (Fig. 5a, b), in line with the lack of changes in the transcriptome and conversion rate analysis (Fig. 2e; Additional file 1: Fig. S8e, f). We did not observe any discernible increase in chromatin accessibility with simultaneous overexpression of Neurog2 + Hmgb2 compared with Neurog2 in EGF + bFGF culture (Fig. 5a). However, we observed a substantial increase in chromatin accessibility after simultaneous overexpression of Neurog2 + Hmgb2 compared with Neurog2 in the bFGF culture (Fig. 5b). This increase in TSS (± 3 kb) accessibility might have been due to at least two mutually non-exclusive mechanisms: (a) widespread TSS opening after Hmgb2 overexpression or (b) lineage-specific changes. Therefore, we analyzed the TSS accessibility of neuronal cell-type-specific genes [58] (Fig. 5c). Whereas we observed increased accessibility of these sites after both Neurog2 and Neurog2 + Hmgb2 overexpression in the EGF + bFGF culture condition, in the bFGF culture condition, the increase in these sites was detectable only after simultaneous overexpression of Neurog2 + Hmgb2 but not Neurog2 alone (Fig. 5c). Interestingly, the TSS opening was comparable between bFGF and EGF + bFGF astrocytes after Neurog2 + Hmgb2 overexpression (Fig. 5c), in line with an increased conversion rate. Next, we wondered whether the Hmgb2-dependent increase in accessibility might be confined to neuronal genes or whether it might also occur in genes specific for other cell lineages. Therefore, we analyzed the dependence of the promoter accessibility of genes identifying ES cells [59, 60], endothelial cells [61–63], and microglial cells [64, 65] on Hmgb2 levels in bFGF culture (Fig. 5d). We found no significant differences in accessibility between the Hmgb2-, Neurog2-, or Neurog2 + Hmgb2-treated astrocytes and the controls, thus indicating that the accessibility change after Neurog2 + Hmgb2 overexpression was specific for neuronal fate.

To identify direct conversion relevant changes in chromatin accessibility dependent on Hmgb2 levels, we determined the significant differentially accessible sites (DASs) after overexpression of Neurog2 and Neurog2 + Hmgb2, compared with CAG-GFP-transduced cells, in the bFGF and EGF + bFGF culture conditions. In the bFGF culture, Neurog2 overexpression resulted in 612 DASs (445 more accessible sites (MASs) and 167 less accessible sites (LASs); Fig. 5e, Additional File 5: Table S4). Combined overexpression of Neurog2 + Hmgb2 in the bFGF culture resulted in 1213 DASs (1062 MASs

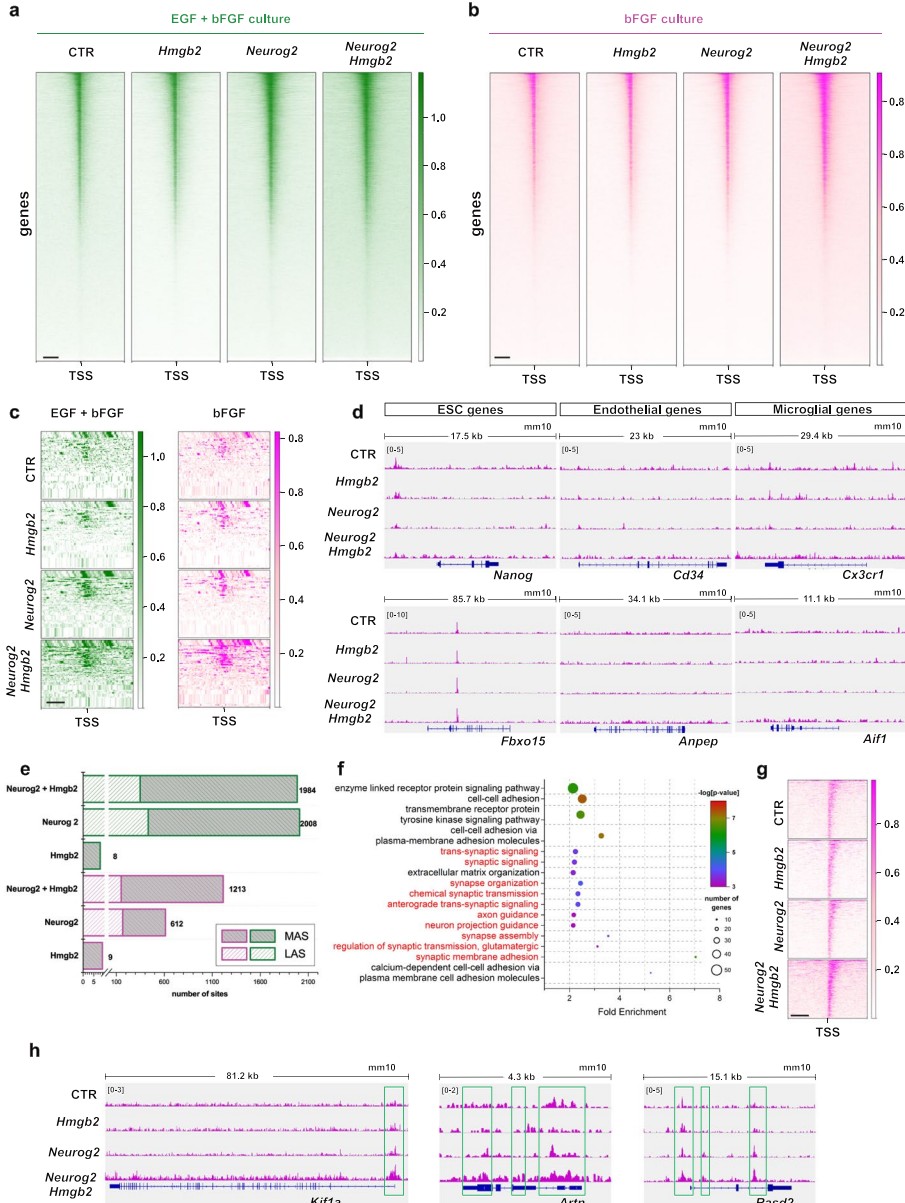

**Fig. 5** Hmgb2 improves the capability of Neurog2 to open promoters of neuronal maturation-associated genes **a,b** Heat maps depicting opening of promoters by Neurog2 and Hmgb2 or their combination in EGF + bFGF (green, **a**) and bFGF (magenta, **b**) culture. Scale: 1 kb (**c**) Heat maps depicting ATAC signals in the promoters of the core neurogenic genes (Fig. 4d) 48 h after Neurog2, Hmgb2, or Neurog2 + Hmgb2 overexpression in EGF + bFGF and bFGF cultures. **d** IGV tracks showing the ATAC signal in the promoters of genes identifying non-neuronal lineages 48 h after Neurog2, Hmgb2, or Neurog2 + Hmgb2 overexpression in bFGF culture. **e** Histogram depicting the number of more (MAS) or less (LAS) accessible sites identified by ATAC 48 h after Neurog2, Hmgb2, or Neurog2 + Hmgb2 overexpression in EGF + bFGF (green) and bFGF (magenta) cultures. **f** Plot depicting enriched GO biological process terms in the promoter set opened by Neurog2 + Hmgb2 in bFGF culture 48 h after viral transduction. Red text represents the GO terms associated with neuronal fate. **g** Heat map showing ATAC signal in the promoters of neuronal maturation-related genes (red in panel **f**) 48 h after Neurog2, Hmgb2, or Neurog2 + Hmgb2 overexpression in bFGF culture. **g** IGV tracks showing the ATAC signal in the promoters of representative genes involved in neuronal maturation 48 h after Neurog2, Hmgb2, or Neurog2 + Hmgb2 overexpression in FGF culture. Green boxes indicate differentially accessible sites

and 151 LASs; Fig. 5e, Additional file 1: Fig. S9a). However, this increase in accessibility did not change the accessibility profile induced by Neurog2 and Neurog2+Hmgb2 in the bFGF culture, because we observed a similar distribution of MAS in the gene bodies, promoters, and intergenic regions (Additional file 1: Fig. S9b, c). Importantly, the Hmgb2-associated increase in MASs was not observed in EGF+bFGF astrocyte culture (Fig. 5e), in agreement with our transcriptome analysis. To reveal the processes influenced by MASs, we analyzed genes associated with these sites (defined as genes within 3 kb upstream and downstream of the MAS) in GO analysis. MASs induced by the simultaneous overexpression of Neurog2+Hmgb2 in the bFGF culture were associated with nervous system development, synaptic membrane adhesion, axon guidance, synapse assembly, and chemical synaptic transmission (Fig. 5f, Additional File 6: Table S5). This finding suggests that Hmgb2 (together with Neurog2) increases the accessibility of genes involved in neuronal maturation. Indeed, the promoters of synapse-associated genes such as *Kif1a* [66, 67], *Artn* [68], and *Rasd2* [69] were closed in the bFGF culture after either control viral transduction or Hmgb2 overexpression (Fig. 5h), in line with the astrocytic fate of these cells. Moreover, Neurog2+Hmgb2 overexpression opened the synapse-associated promoters to a significantly greater extent than Neurog2 alone (Fig. 5g, h). We then asked whether the chromatin opening state of all or only a subset of Neurog2-induced maturation genes depended on the expression of Hmgb2. Therefore, we compared the MASs induced by Neurog2 in the two conversion prone conditions (overexpression of Neurog2 in EGF+bFGF and overexpression of Neurog2+Hmgb2 in bFGF culture) with MASs induced by Neurog2 in the conversion resistant condition (overexpression of Neurog2 in bFGF culture). We identified 395 MASs commonly induced in both conversion prone conditions (Fig. 6a). These MASs were enriched in processes associated with synapse formation (GO biological processes such as nervous system development, synaptic organization, trans-synaptic signals, potassium transport, and synaptic membrane adhesion; Fig. 6b, Additional File 7: Table S6a). Importantly, the increase in the accessibility of these synapse-associated loci correlated with the increased expression of these genes after Neurog2+Hmgb2 overexpression in bFGF culture (Additional file 1: Fig. S10a, b). However, we also observed 268 MASs induced by Neurog2 in all three conditions (Fig. 6a) that were enriched in synaptic processes (Fig. 6c, Additional File 7: Table S6b). Therefore, these data suggested that the chromatin containing only a subset of genes associated with neuronal maturation was dependent on Hmgb2. However, the accessibility of these genes appeared to be instrumental for direct conversion.

Together, our data supported a model in which Hmgb2 fosters the establishment of the full neurogenic program by increasing the accessibility and consequently the expression of neuronal maturation genes, thus leading to improved neuronal maturation.

### Hmgb2-dependent chromatin sites contain both E-boxes and Pou factor binding sites important for neuronal maturation

HMG proteins play a major role in controlling gene expression by increasing chromatin accessibility [70–72]. Therefore, we sought to identify the potential TF binding motifs enriched in the Hmgb2-dependent set of MASs (395 sites in Fig. 6a). To do so, we performed de novo motif enrichment analysis using BaMMmotif software. Motifs

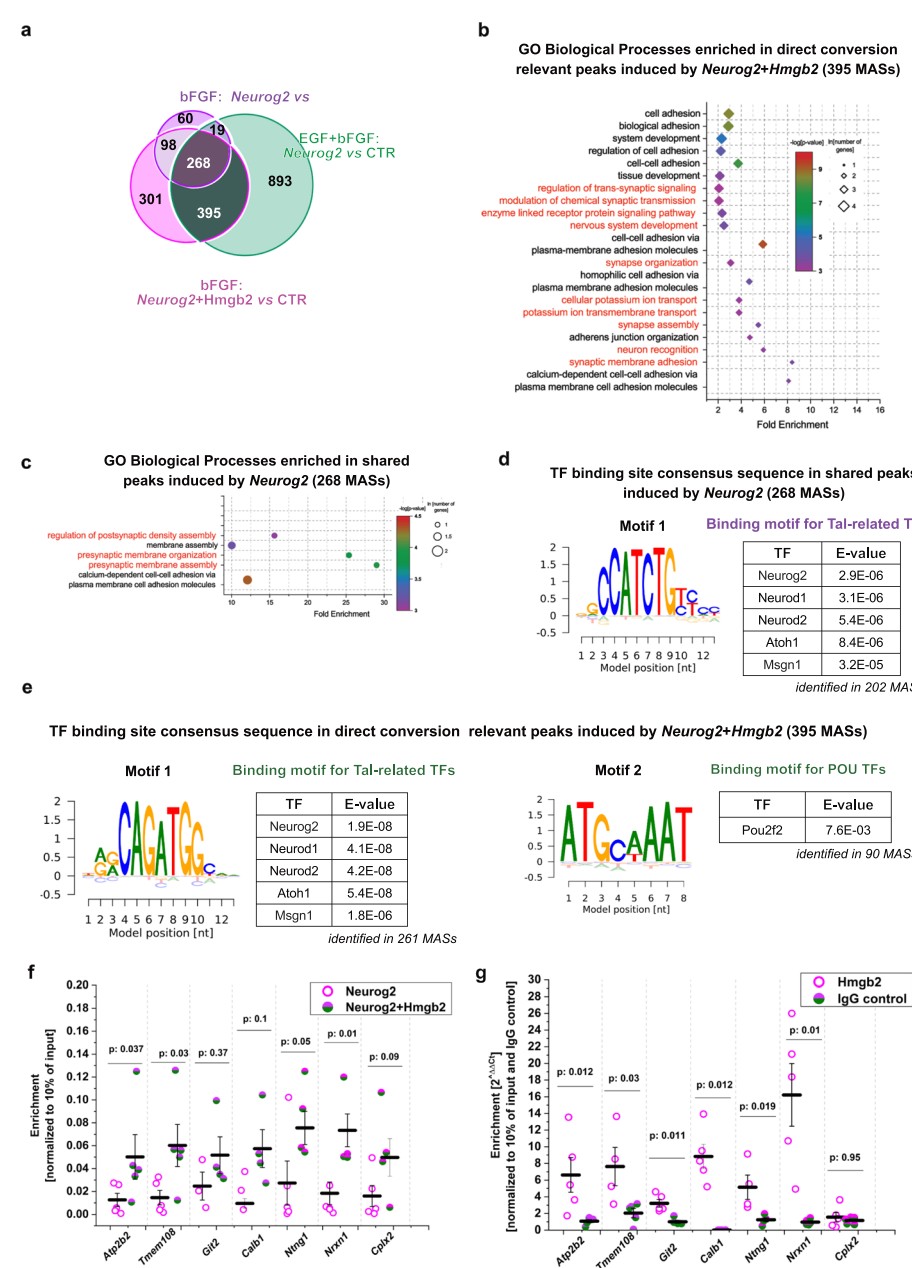

**Fig. 6** Hmgb2-dependent promoters contain an E-box and Pou2f2 factor binding motif **a** Venn diagram illustrating the overlap in ATAC signals for MASs after Neurog2 overexpression in EGF + bFGF and bFGF cultures, with MASs induced by Neurog2 and Hmgb2 overexpression in bFGF culture 48 h after viral transduction. **b**, **c** Plots depicting enriched GO biological process terms in 395 peak set MASs in panel **a** (**b**) and 268 peak set MASs in panel **a** (**c**). Red text represents the GO terms associated with neuronal fate. **d, e** Transcription factor consensus sequences identified in 268 peak set MASs in panel **a** (**d**) and 395 peak set MASs in panel **a** (**e**), identified with de novo motif analysis. The motif image from the BaMM web server shows the likelihood of each nucleotide at each motif position. The color intensity reflects the probability, with darker colors indicating higher probabilities. Tables show transcription factors binding these motifs. **f, g** Dot plots show the enrichment of selected promoters from genes associated with the GO term "modulation of chemical synaptic transmission" (Fig. 6b), compared to 10% input, using the FLAG antibody (targeting Neurog2-FLAG) and the Hmgb2 antibody in bFGF culture. Data are shown as median ± IQR; single dots represent independent biological replicates. Significance was tested with two-tailed Mann-Whitney test. Abbreviations: MAS, more accessible site; TF, transcription factor

containing the consensus binding sequence of the Tal-associated TF family (Neurod1, Neurog2, Neurod2, Atoh1, and Msgn1) were enriched in Hmgb2-dependent set of MASs (Fig. 6d, Additional File 8: Table S7a). In addition, we identified the motif that best matched the consensus sequence of the TF family of POU domain factors, such as Pou2f2 (Fig. 6e, Additional File 8: Table S7b). Pou2f2 is a direct Neurog2 target [73] and has been reported to be involved in the implementation of proper neuronal identity [74, 75]. This finding suggested that in the bFGF culture, some of the E-box motif sites bound by Neurog2 (Tal related factors) were inaccessible, but with the addition of Hmgb2, these sites became accessible, thereby increasing Neurog2-binding and enhancing reprogramming efficiency. To validate this concept, we focused on genes associated with the GO term "modulation of chemical synaptic transmission" (Fig. 6b) and examined NEUROG2 binding to their promoters in bFGF culture, as a function of HMGB2 levels. Our analysis revealed that NEUROG2 binding to the promoters of these genes was significantly enhanced when Neurog2 and Hmgb2 were co-expressed (Fig. 6f). Additionally, we demonstrated direct HMGB2 binding to most of these promoters (Fig. 6g), suggesting that HMGB2 binding increases the promoters' accessibility for NEUROG2. Interestingly, some promoters, such as the Cplx2 promoter, were not directly bound by HMGB2 (Fig. 6g). However, NEUROG2 binding to these promoters still depended on HMGB2 levels (Fig. 6f). This indicates the involvement of alternative mechanisms, beyond direct co-binding, in facilitating NEUROG2's interaction with such promoters. We next investigated MASs with consensus binding sequences for both Tal-associated factors (Neurog2) and POU domain factors. We identified that 56 of 395 MASs contained binding motifs for both TF families (Additional file 1: Fig. S10c) and were associated with neuronal maturation (GO processes: regulating actin filaments assembly, chemotaxis, and potassium ion transport; Additional file 1: Fig. S10d and Additional File 7: Table S6c), including the Robo-Slit pathway. Robo-Slit pathway has been reported to regulate not only axonal pathfinding but also neuronal maturation [76]. Moreover, we observed enrichment in genes associated with the negative regulation of proliferation, thus possibly improving the terminal differentiation of converted cells. Interestingly, de novo motif analysis of the common 268 Neurog2-induced MASs identified the binding motif of the TF family of Tal-associated factors, but not of the POU domain factors (Fig. 6d). These data suggested that Hmgb2 levels set the lineage barrier by controlling the accessibility of both the direct Neurog2 targets and targets of TFs downstream of Neurog2, such as Pou3f2 or Neurod.

To directly test the importance of Hmgb2 in neuronal maturation, we analyzed the neurite complexity of the converted neurons in the conversion prone cultures (overexpression of Neurog2 in EGF + bFGF and overexpression of Neurog2 + Hmgb2 in bFGF culture) and the conversion resistant culture (overexpression of Neurog2 in bFGF culture) in induced neurons with Sholl analysis 7 days after viral transduction (Fig. 7a). Indeed, Neurog2-induced neurons in the bFGF culture showed fewer intersections than the Neurog2-induced neurons in the EGF + bFGF culture (Fig. 7b, c). Lower neurite complexity is indicative of less mature neurons. The complexity of neurites in neurons generated from bFGF astrocytes by the combined overexpression of Neurog2 and Hmgb2 increased compared to overexpression of Neurog2 only. These

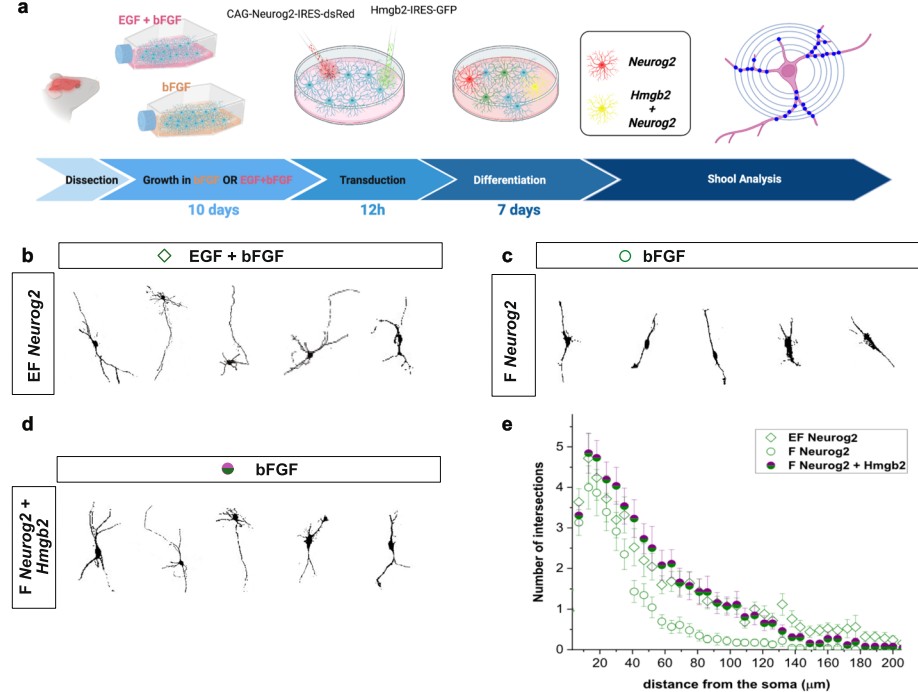

**Fig. 7** Hmgb2 and Neurog2 overexpression increases complexity of iN **a** Scheme depicting the experimental paradigm used for Sholl analysis. **b–d** Representative thresholded images of neuronal cells used for Sholl analysis. **e** Sholl analysis of induced neurons by concurrent overexpression of Neurog2 and Neurog2 + Hmgb2 in EGF and EGF + bFGF culture 7 days after viral transduction. Abbreviations: MAS, more accessible site; TF, transcription factor

converted neurons were indistinguishable from those generated by overexpression of Neurog2 in the EGF + bFGF-cultured astrocytes (Fig. 7b, c).

## Discussion

The establishment of neuronal identity during direct astrocyte to neuron conversion is achieved in very different environmental context from that of the bona fide neurogenesis occurring during embryonic development or in adult brain neurogenic niches [46, 48]. This includes not only the different starting populations [46] but also the unique signaling milieus [77–79]. The growth factors released after injury regulate the conversion process, including neuronal maturation and neural circuit repair. Here, we presented a novel in vitro system to study the influence of growth factors on fate conversion. Using this system, we showed that EGF, potentially provided by the injured environment, is necessary for efficient neuronal conversion and proper maturation via the regulation of the chromatin binding protein Hmgb2. In combination with several different neurogenic fate determinants, Hmgb2 is capable of inducing the full neurogenic program, as indicated by Hmgb2 gain and loss of function experiments. Our model suggests that exposure to EGF is essential for conversion in the injury-induced environment. In contrast, the prolonged elevation of bFGF levels may contribute to the low conversion rate, as FGF signaling alone promotes processes related to neurogenesis and neuronal fate in astrocytes during Neurog2-mediated conversion. This finding is in line with reports that the FGF promotes neurogenesis [80–82], although the neuronal subtypes generated in such

context differ [82]. Importantly, the chromatin states in direct conversion and during embryonic neurogenesis may differ: the chromatin states during neurogenesis require fewer re-arrangements in embryonic development, because large numbers of neurogenic gene loci in radial glial cells, the neuronal stem cells of the developing CNS, are already in an open configuration [28, 83]. Interestingly, genes involved in synapse formation and neuronal maturation are already in an active chromatin state without detectable gene expression in both radial glia and committed neuronal progenitors [28, 84], thus implying the existence of an active inhibitory mechanism keeping the progenitor state primed toward neurogenesis and preventing their premature differentiation. Importantly, Hmgb2 opens the loci of these classes of genes during astrocyte to neuron conversion, thus supporting the concept that overexpression of Neurog2 + Hmgb2 endows postnatal astrocytes with some stem cell features. This concept is also in line with the expression of Hmgb2 during activation of quiescent neural stem cells in the adult brain [50] and its role in adult neurogenesis [85]. However, we did observe immediate expression of synaptic genes in postnatal astrocytes without the maintenance of these primed neuronal states, thus suggesting that the mechanisms preventing premature differentiation operating in the neuronal stem cells are not established during astrocyte to neuron conversion. This possibility reinforces the concept that direct neuronal conversion does not fully recapitulate the developmental trajectory underlying neuronal differentiation [42, 45]. Instead, the overexpression of reprogramming factors induces early re-arrangements of chromatin along with changes in gene expression. However, during late morphological and functional maturation stages of the induced neurons, changes in chromatin are negligible [26]. Moreover, in our in vitro system, we did not observe any changes in astrocyte proliferation due to the overexpression of Hmgb2 alone or in combination with different neurogenic TFs, thus further limiting the spectrum of neural stem cell features induced in the postnatal astrocytes. Interestingly, Hmgb2 induces similar chromatin changes in postnatal astrocytes to the HMG group protein A2, a different HMG-box-containing family member. These chromatin changes are sufficient to prolong the neurogenic phase during cortical development and lead to the generation of new postnatal neurons [86]. During this period, progenitors normally generate glial cells, thus potentially implicating similar mechanisms in the Hmga2-mediated extension of neurogenic period and the Hmgb2-mediated direct astrocyte to neuron conversion. Because Hmga2 is associated with Polycomb signaling [87], testing whether the same system would be operational during the Hmgb2-dependent conversion should prove interesting, because Ezh2 maintains the lineage barriers during fibroblast to neuron conversion [88]. Both Hmgb2 and Hmga2 bind AT-rich DNA segments with little to no sequence specificity [71, 89]. Nevertheless, we observed highly specific Hmgb2-dependent opening of chromatin containing late neuronal maturation genes, thus prompting questions regarding HMG protein binding specificity. This specificity could be provided by an interacting protein, e.g., neurogenic TF Neurog2, because we observed an enrichment of the typical E-box binding sequence in the promoters when Hmgb2 was overexpressed in astrocytes. However, our findings did not reveal a direct interaction of Hmgb2 with Neurog2 via WB or mass spectrometry (data not shown), thus making this scenario unlikely. An alternative explanation may be that Hmgb2 stabilizes the regulatory loops (transactivation domains, TADs) involved in the expression of synaptic genes.

The regulatory roles of such domains have been demonstrated for neurogenesis downstream of Neurog2 during embryonal cerebral cortex development [27]. Moreover, both Hmgb2 and Hmga2 have been implicated in TAD establishment [90, 91]. The stabilization of regulatory loops induced by Neurog2 may indeed provide a mechanistic explanation for the Hmgb2-dependent opening of chromatin regions containing the Neurog2 binding E-boxes. These data further challenge the common belief that Neurog2 is a pioneer TF. In contrast to the on-target pioneering function of Ascl1 during reprogramming [25, 26], in fibroblast to neuron conversion, Neurog2 requires additional factors, such as forskolin and dorsomorphin or Sox4, that are necessary for not only late neuronal maturation but also the induction of early reprogramming changes [30, 73]. We demonstrated that, at least in the case of astrocyte to neuron conversion, Neurog2 function is dependent on Hmgb2. Because Hmgb2 increases the accessibility of various sites, including the binding motif of the Neurog2 target Pou2f2 [27], our data suggested that Neurog2 must open the chromatin of maturation genes that are transcriptionally regulated by direct Neurog2 targets. Our study provides mechanistic insights into previously described improvements in neuronal reprogramming with the infusion of EGF and FGF [10]. Interestingly, EGF and FGF exhibit different temporal dynamics post-injury, with a very narrow expression window and a presumably diminished activity window of EGF [34]. This window correlates with the expression of Hmgb2, thus suggesting that prolonged expression of either EGF or Hmgb2 after TBI might be important in the success of neuronal replacement therapies. Furthermore, our model may also explain the lower direct conversion rates induced by Neurog2 in some starting cellular populations, such as oligodendrocyte precursor cells [29], in which the promoters might not yet be open. Similarly, such multilevel control is compatible with the ability of Neurog2 to induce different neuronal subtypes or maturation stages in different, permissive starting cells [30, 44, 92, 93], given that maturation loci defining the neuronal subtype could be differentially accessible for Neurog2 direct targets.

Interestingly, the overexpression of Neurog2 in bFGF-grown astrocytes induced not only a partial neurogenic program but also additional transcriptional programs associated with alternative fates, such as cartilage formation and immune cell differentiation. The induction of alternative fates or a failure to repress the original fate can lead to abortive conversion and concomitant death of reprogrammed cells [23], thereby possibly mechanistically explaining the lower Neurog2-mediated conversion efficiency in the bFGF culture. Because Hmgb2 overexpression does not specifically repress the astrocytic fate, yet significantly improves the conversion efficiency, the abortive direct conversion is unlikely to explain the lower efficiency in direct conversion. Interestingly, we did not observe Hmgb2-dependent opening of regions associated with alternative fate genes, thus supporting the idea that alternative fate induction is independent of the Hmgb2-induced changes in chromatin states. Hmgb2-dependent changes in the transcription rate [94], RNA stability or RNA splicing could account for the enrichment of alternative fates observed in mRNA analysis, because Hmgb2 has been proposed to have an RNA-binding domain [89]. Importantly, we observed changes in chromatin opening for only genes associated with the neurogenic lineage.

## Conclusions

Together, our results provide a mechanistic framework for translating environmental signals into a specific program involved in neuronal maturation downstream of the neurogenic fate determinants via chromatin modification. Interestingly, this aspect of neuronal reprogramming is the least understood and stands to be further improved, particularly in vivo.

## Methods

### Experimental animals

Experiments were conducted on both female and male animals, which were either wild types (C57BL/6 J mice) or transgenic Hmgb2−/− animals on a C57BL/6 background [95]. The Hmgb2−/− mice do not show gross phenotypical abnormalities and do not differ to wild-type siblings (Ronfani et al., 2001). For all in vitro experiments, animals at postnatal stage P5-P6 were used. Injuries were done in adult 8–10-week-old animals. Animals were kept under standard conditions with access to water and food ad libitum. All animal experimental procedures were performed in accordance with the German and European Union guidelines and were approved by the Institutional Animal Care and Use Committee (IACUC) and the Government of Upper Bavaria under license number: AZ 55.2–1–54–2532-171–2011 and AZ 55.2–1–54–2532-150–11. All efforts were made to minimize animal suffering and to reduce the number of animals used.

### Stab wound injury and EGF infusion

Prior to every surgery, mice were deeply anesthetised by intra-peritoneal injection of sleep solution (Medetomidin (0.5 mg/kg) / Midazolam (5 mg/kg) / Fentanyl (0.05 mg/kg)) complemented by local lidocaine application (20 mg/g). After the injection of the anesthesia, mice were checked for pain reactions by pinching their tail and toes. Stab wound injury was performed in the somatosensory cortex, as previously described [29, 96]. The following coordinates relative to Bregma were used: medio-lateral: 1.0 µm; rostro-caudal: −1.2 µm to −2.2 µm; dorso-ventral: −0.6 µm. Anesthesia was antagonized with a subcutaneous injection of awake solution (Atipamezol (2.5 mg/kg) / Flumazenil (0.5 mg/kg) / Buprenorphin (0.1 mg/kg)), and the mice were kept on a pre-warmed pad until they were awake and recovered from the surgery.

For EGF application, 500 ng of EGF or artificial cerebrospinal fluid was delivered on the brain surface using microinjection system. Animals were perfused 24 h later and analyzed for HMGB2 expression.

### Perfusion and tissue section preparation

Prior to perfusion, animals were deeply anesthetized with overdoses of cocktail of ketamine (100 mg/kg) / xylazine (10 mg/kg). Subsequently, they were transcardially perfused first with cold PBS, followed by fresh ice-cold 4% PFA in PBS for 20 min. The brain was then removed from the skull, post-fixed in the same fixative overnight at 4 °C, cryoprotected in 30% sucrose and cut at the cryostat at 40-µm-thick sections.

### Preparation of PDL-coated glass coverslips

Glass coverslips were washed first with acetone and boiled for 30 min in ethanol containing 0.7% (v/v) HCl. After two washing steps with 100% ethanol, coverslips were dried at RT and autoclaved for 2 h at 180 °C. Coverslips were washed with D-PBS and coated with poly-D-lysine (PDL, 0.02 mg/ml) solution for at least 2 h at 37 °C. Following coating, coverslips were washed three times with autoclaved ultrapure water, dried in the laminar flow, and stored at 4 °C until needed.

### Primary culture of postnatal cortical astroglial cells

Postnatal cortical astroglia were isolated and cultured as described previously [97]. Following decapitation of postnatal (P5-P6) wild-type C57BL/6 J mice, the skin and the skull were removed, and the brain was extracted avoiding any tissue damage and placed into the 10 mM HEPES solution for dissection. After separating the two hemispheres, the meninges were removed and white matter of cerebral cortex was dissected using fine forceps and collected in a tube with astrocyte medium (fetal calf serum-FCS (10% (v/v)); horse serum-HS (5% (v/v)); glucose (3.5 mM); B27 supplement; Penicillin/Streptomycin (100 I.U/ml Pen and 100 µg/ml Strep) in DMEM/F12+GlutaMAX). The tissue was mechanical dissociated with a 5-ml pipette and placed into uncoated plastic flasks for cell expansion in astrocyte medium supplemented with the two growth factors EGF (10 ng/ml)+bFGF (10 ng/ml each) or with bFGF (10 ng/ml) only as specified for each experiment. After 4–5 days, the medium was exchanged and supplied with the fresh growth factors. After 10 days of culturing, cultured cells were rinsed with DPBS and contaminating oligodendrocyte precursor cells were removed by brusquely shaking the culture flasks several times. Astroglial cells were then detached from the flask by trypsinization and seeded onto poly-D-lysine (PDL)-coated glass coverslips at a density of $8 \times 10^4$ cells per well in a 24-well plate with astrocyte medium for immunohistochemical analysis. For the ATAC-seq and RNA-seq experiments, cells were plated in T75 flasks with a seeding density of $3 \times 10^6$ cells per flask. Two to four hours after seeding, the cells were transduced with different retroviral vectors in a ratio of 1 µl virus per 1 ml medium to prevent virus toxicity. Astrocyte medium was changed 12–18 h after viral transduction to differentiation medium (glucose (3.5 mM); B27 supplement; Penicillin/Streptomycin (100 I.U/ml Pen and 100 µg/ml Strep) in DMEM/F12+GlutaMAX) containing neither EGF nor bFGF up to the immunocytochemical analysis timepoint. The cells were cultured as indicated in each experiment. Cells were fixed in cold 4% PFA for 20 min and rinsed with cold D-PBS before immunocytochemical analysis.

For the ATAC-seq and RNA-seq experiments, the cells were kept in the astrocyte medium and collected 48 h after viral transduction. Astrocytes were detached from the flask by trypsinization, prepared for the FACS and sorted for the following ATAC-seq and RNA-seq experiments according to the fluorophore expression.

The astroglial cultures from the Hmgb2−/− transgenic animals were prepared as described above; however, the cortical tissue from each animal was kept separately and placed into the small T25 flask. In addition, the tips of the tails were used for genotyping as described in [95]. The cultures from Hmgb2−/− transgenic mice were grown only in the double growth factor condition containing EGF+bFGF.

**Immunocytochemistry and immunohistochemistry**

Immunostaining was performed on cell culture samples or free-floating brain sections. Specimens were treated with blocking buffer (0.5% Triton-X-100; 10% normal goat serum (NGS) in D-PBS) to reduce non-specific binding. The same buffer was used to dilute the primary antibodies. The specimens were incubated with the primary antibody mixture overnight at 4 °C (brain tissue) or for 2 h at RT (cell culture samples), followed by $3 \times 10$ min washing steps with PBS. In order to visualize primary antibody binding, samples were exposed to appropriate species and/or subclass-specific secondary antibodies conjugated to Alexa Fluor 488, 546 or 647 (Invitrogen) for about 90 min at RT protected from light. Secondary antibodies were diluted 1:1000 in blocking buffer. Nuclei were visualized with DAPI (4',6-diamidino-2-phenylindole) that was added to the mix of secondary antibodies. Following extensive washing steps with PBS, coverslips or sections were mounted with Aqua Poly/Mount (Polysciences) and imaged.

The following primary antibodies were used: Chick-anti-GFP (Aves Lab, GFP-120; 1:1000); Rabbit-anti-RFP (Rockland, 600–401-379; 1:500); Mouse IgG1-anti-GFAP (Sigma-Aldrich, G3893; 1:500); Rabbit-anti-GFAP (DakoCytomation, Z0334; 1:1000); Mouse IgG1κ-anti-S100β (Sigma-Aldrich, S2644; 1:500); Rabbit-anti-OLIG2 (Thermo Fischer, AB9610; 1:500); Mouse IgG2a-anti-αSMA (Sigma-Aldrich, A2547; 1:400); Rabbit-anti-Ki67 (Abcam, 15,580; 1:200); Rat-anti-Ki67 (DakoCytomation, M7249; 1:200); Rabbit-anti-PH3 (Ser10) (Thermo Fischer, 06–570; 1:200); Guinea pig-anti-DCX (Thermo Fischer, AB-2253; 1:1000); Mouse IgG2b-anti-β-III-TUBULIN (Sigma-Aldrich, T8660; 1:500); Mouse IgG1-anti-NEUN (Chemicon, MAB 377; 1:250); Rabbit-anti-HMGB2 (Abcam, ab67282; 1:1000); Mouse IgG2aκ-anti-HMGB2 (Sigma-Aldrich, 07173-3E5; 1:500); Mouse IgG2aκ-anti-HMGB2 antibody requires thermal (15 min at 95 °C) antigen retrieval using the citrate buffer (10 mM; pH 6). Primary antibody binding was revealed using class-specific secondary antibody coupled to Alexa fluorophore (Invitrogen, Germany). All secondary antibodies were used at dilution 1:1000.

**Image acquisition and quantifications**

Immunostainings were analyzed with a fluorescent Microscope Axio Imager M2m (Zeiss) using the ZEN software (Zeiss) with a $\times 20$ or $\times 40$ objective. Fluorescent-labelled sections were photographed with a FV1000 confocal laser-scanning microscope (Olympus), using the FW10-ASW 4.0 software (Olympus). The quantifications of in vitro cultured cells were performed using the ZEN software (Zeiss) analyzing at least 25 randomly taken pictures per coverslip depending on the number of transduced cells. In total, 100–200 retroviral vector-transduced cells were quantified from randomly chosen fields on a single coverslip. Three coverslips in each experiment (biological replicate) were analyzed. The number of experiments is indicated in the corresponding figure. The number of induced neurons was expressed as a percentage out of all transduced cells.

To analyze the number of apoptotic cells, between 350 and 550 DAPI labelled cells were counted from 5 randomly selected fields on one coverslip.

In the reprogramming experiments of the astrocytes isolated from Hmgb2 + / + , Hmgb2 + / − , and Hmgb2 − / − animals, each of the single animals was considered as

a biological replicate and at least 3 coverslips were counted per animal. We analyzed in total 6 litters containing wild-type, heterozygous or homozygous littermates.

Western blots using the Fiji software were previously described [98]. All lanes of interest were outlined using the rectangular selection tool and the signal intensity of each band was calculated by determining the area under the peak. The measurements of the corresponding α- ACTIN bands were used to normalize the amount of proteins loaded on the gel.

The intensity of HMGB2 and HMGB1 stainings was performed on single optical sections using FiJi plug-in.

### Sholl analysis

We analyzed only DCX-positive cells 7 days after viral transduction. Single cells were isolated and subjected to Sholl analysis using the ImageJ plug-in "Sholl Analysis." We used the following parameters: starting radius 5 μm; ending radius 500 μm; radius step size 5 μm. The number of crossings per cell was visualized and analyzed using Origin.

### FACS analysis and sorting

Astrocytes were collected by trypsinization 48 h after retroviral transduction, washed, resuspended in DPBS and analyzed using a FACS Aria II instrument (BD Biosciences) in the FACSFlowTM medium. Debris and aggregated cells were gated out by forward-scatter area (FSC-A) and side-scatter area (SSC-A). Forward-scatter area (FSC-A) vs. forward-scatter width (FSC-W) was used to discriminate doublets from single cells. To set the gates for the sorting, untransduced astrocytes were recorded. Sorted cells were collected in DPBS, counted and divided into two batches: 50,000 cells were immediately processed for ATAC-seq and the remaining cells were collected for RNA-seq library preparation.

### ATAC-sequencing

Assay for Transposase Accessible Chromatin with high-throughput sequencing (ATAC-seq), a method to detect genome-wide chromatin accessibility, was performed following the published protocol [99, 100]. Briefly, right after the FACS sorting, 50,000 cells were lysed, the nuclei were extracted and resuspended with the transposase reaction mix (25 μl 2 × TD buffer (Illumina); 2.5 μl Transposase (Illumina); 22.5 μl nuclease-free water), followed by transposition reaction for 30 min at 37 °C. To stop the transposition reaction, samples were purified using a Qiagen MinElute PCR (Qiagen) purification kit according to the manufacturer instructions. Open chromatin fragments were first amplified for 5 cycles and then for additional 7–8 cycles, as determined by RT-qPCR, using the combination of primer Ad1_noMX (5′ AATGATACGGCGACCACCGAGATCTAC ACTCGTCGGCAGCGTCAGATGTG 3′) and the Nextera Index Kit (Illumina) primer N701-N706. Libraries were purified using a Qiagen MinElute PCR purification kit (Qiagen), and their quality was assessed using the Bioanalyzer High-Sensitivity DNA kit (Agilent) according to the manufacturer's instructions. The concentration of each library was measured by Qubit using the provided protocol. Libraries were pooled for sequencing and the pool contained 20 ng of each library. Prior to sequencing, pooled libraries were additionally purified with AMPure beads (ratio 1:1) to remove contaminating

primer dimers and quantified using Qubit and the Bioanalyzer High-Sensitivity DNA kit (Agilent). Fifty-base pair paired-end deep sequencing was carried out on HiSeq 4000 (Illumina).

### ATAC-sequencing analysis

For the analysis of bulk ATAC-seq data, we followed the Harvard FAS Informatics ATAC-seq guidelines. The quality of raw FASTQ reads was checked using FastQC (Version 0.11.9). The low-quality read (< 20 bp) and adapter sequences were trimmed by Cutadapt (Version 4.0). The trimmed reads were mapped to the mouse reference genome (mm10) by using Bowtie2 (parameter: –very-sensitive -X 1000 –dovetail). Samtools were then used to convert and sort the sam files into bam files. Peak calling step was performed with Genrich for each sample separately to identify accessible regions. Genrich peak caller has a mode (-j) assigned to ATAC-Seq analysis mode and allows running all of the post-alignment steps via peak-calling with one command. Mitochondrial reads and PCR duplicates were removed by -e chrM and -r argument respectively. To generate count table matrix for differential analysis bam2counts (intePareto R-based package) was used to count reads fall into specific genomic positions by importing all the bam files and merging all the bed files into one (importing GenomicRanges and GenomicAlignments libraries). DESeq2 (version 1.26.0) was used for differential accessibility analysis of the count data. The relatively more open and closed sites are called MAS and LAS respectively (fold change (FC) > 2 and adjusted $P$-value < 0.05), and the annotation of these sites were performed using R-based packages Chip-seeker (TSS ± 3.0 kb) (version 1.28.3). For visualization, the bamcovage deeptools (version 3.5.1) were used to normalize the data by importing the scaling factor from DESeq2 (version 1.36.0). The normalized bigwig files used to visualize the coverage using deeptools and samtools. These bigwig files were loaded into the IGV tool to visualize the peak at the gene level. The Venn diagrams were made using the BioVenn web application tool. The Gorilla tool was used to generate the GO Biological processes, with a cut-off of enrichment > 2 and $p$-value of < 0.01.

### Motif analysis

BaMMmotif (https://bammmotif.soedinglab.org/home/) was used to perform de novo motif enrichment analysis by providing MASs fasta sequence [101] as input and all detected accessible sites fasta sequences as background using default parameters. We selected the motifs with an AvRec score above 0.5 as candidates for further analysis. The mouse database HOCOMOCO v11 was used for motif annotation, and the most significant transcription factors matching the motif with e-values below 0.001 were considered as potential binders.

### Preparation of libraries for RNA-sequencing

Sorted cells were resuspended in 100 µl extraction buffer of the PicoPureTM RNA isolation kit (Thermo Fischer Scientific), and the RNA was extracted according to the manufacturer's instructions. The Agilent 2100 Bioanalyzer was used to assess RNA quality and concentration. For the RNA-seq library preparation, only high-quality RNA with RIN values > 8 were used. cDNA was synthesized from 10 ng of total RNA using SMART-Seq v4 Ultra Low Input RNA Kit (Takara Bio), according to the manufacturer's instructions.

The total number of amplification cycles was determined by RT-qPCR side reaction according to the manufacturer's instruction. PCR-amplified cDNA was purified by immobilization on AMPure XP beads. Prior to generating the final library for sequencing, the Covaris AFA system was used to perform cDNA shearing in Covaris microtubes (microTUBE AFA Fiber Pre-Slit Snap-Cap $6 \times 16$ mm), resulting in 200–500-bp-long cDNA fragments that were subsequently purified by ethanol precipitation. Prior to library preparation using the MicroPlex Library Preparation kit v2 (Diagenode) according to the user manual, the quality and concentration of the sheared cDNA were assessed using an Agilent 2100 Bioanalyzer. Final libraries were evaluated using an Agilent 2100 Bioanalyzer, and the concentration was measured with Qubit Fluorometer (Thermo Fisher Scientific). The uniquely barcoded libraries were multiplexed onto one lane and 100-bp paired-end deep sequencing was carried out at the HiSeq 4000 (Illumina) generating $\sim 20$ million reads per sample.

### Transcriptome data analysis (Bulk RNA Seq)

The raw paired-end FASTQ files were mapped to the mouse reference genome (mm10) using STAR RNA-seq aligner (version 2.7.2b). Aligned reads in the BAM files were then quantified by HTSeq-count (Version 0.9.1) based on annotation file GENCODE Release M25 (GRCm38.p6). The gene-level count matrix was imported into the R/Bioconductor package DESeq2 (version 1.26.0) for normalization and differential expression with FC > 2, adjusted *P*-value < 0.05. Venn diagrams were created using the web application BioVenn tool, and heatmaps were generated using gplots and RColorBrewer R-based/ Bioconductor tools. For GO enrichment analysis of the assigned set of genes, we used the GOrilla tool by providing background genes. The enriched GO term (biological processes) possessing enrichment > 2, containing at least 1% of the input genes, and *p*-value specified in the figure legend were visualized using Origin.

### Protein isolation and Western blot

Postnatal cortical astroglia were isolated and cultured as described above. After 10 days of culturing with growth factors EGF + bFGF or bFGF, cells were detached from the flask by trypsinization, washed and counted. $0.5 \times 10^6$ cells were lysed in RIPA buffer containing cOmplete Protease Inhibitor cocktail (Roche). Protein extraction and Western blotting are performed as previously described [102]. The following antibodies were used: Rabbit-anti-HMGB2 (Abcam, ab67282; 1:5000), Mouse-anti-ACTIN (Millipore, MAB1501; 1:10,000), HRP-coupled anti-mouse IgG1 (GE Healthcare, NA931; 1:20,000), and HRP-coupled anti-rabbit IgG (Jackson ImmunoResearch,111–036–045; 1:20,000).

### Quantitative PCR (qPCR)

RNA extraction was performed using the EXTRACTME® TOTAL RNA MICROSPIN KIT in accordance with the manufacturer's protocol. To eliminate potential genomic DNA contamination, the optional DNA removal step was included using DNase I (Sigma-Aldrich). Reverse transcription (RT) was carried out with oligo(dT)18 and random hexamer primers, following the instructions provided in the Maxima First Strand cDNA Synthesis Kit for RT-qPCR. Quantitative PCR (qPCR) was conducted using gene-specific, intron-spanning primers (Additional File 9: Table S8) on the QuantStudio™ 6

Flex Real-Time PCR System, employing the PowerUP™ SYBR Green Master Mix. All experiments were performed in four biological replicates, and data analysis was conducted using Origin software.

### ChIP-qPCR analysis

Astrocytes cultured in bFGF medium were transduced with either a retrovirus expressing FLAG-Neurog2 or a combination of two retroviruses expressing FLAG-Neurog2 and HMGB2. Immunoprecipitation was conducted following a previously established protocol [103]. Forty-eight hours after transduction, 1.5 million astrocytes were cross-linked with 1% formaldehyde for 10 min at room temperature, then lysed in 100 µl of Buffer-B (50 mM Tris–HCl, pH 8.0, 10 mM EDTA, 1% SDS, and $1 \times$ protease inhibitors from Roche). Chromatin was fragmented using a Covaris S220 system under the following settings: 4 °C, duty cycle 2%, peak incident power 105 W, and 200 cycles per burst. This process generated DNA fragments ranging between 200 and 500 base pairs. The lysates were centrifuged at $12,000 \times g$ for 10 min at 4 °C, and the resulting supernatant was diluted with 900 µl of Buffer-A (10 mM Tris–HCl, pH 7.5, 1 mM EDTA, 0.5 mM EGTA, 1% Triton X-100, 0.1% SDS, 0.1% sodium deoxycholate, 140 mM NaCl, and $1 \times$ protease inhibitors). A portion of the lysate (10%) was set aside as an input control, while the remaining chromatin was incubated for 4 h at 4 °C on a rotating wheel with 6 µg of antibody. For FLAG-Neurog2, the anti-FLAG M2 antibody (Sigma, F1804) was used, while HMGB2 was targeted with anti-HMGB2 (Abcam, ab67282). Antibodies were conjugated to 10 µl of Protein G Dynabeads (Thermo Fisher Scientific). For the input controls, 10% of the chromatin was processed identically, except no antibody was added. The antibody-bound beads were washed four times with Buffer-A and once with Buffer-C (10 mM Tris–HCl, pH 8.0, 10 mM EDTA). Immunocomplexes were eluted in 100 µl of elution buffer (50 mM Tris–HCl, pH 8.0, 10 mM EDTA, 1% SDS) by incubating at 65 °C for 20 min. Cross-links were reversed by incubating the eluates overnight at 65 °C. Afterward, 100 µl of TE buffer (10 mM Tris–HCl, pH 8.0, 1 mM EDTA) was added to the samples. RNA was digested with 4 µl of RNase A (10 mg/ml) for 1 h at 37 °C, followed by protein digestion with 4 µl of Proteinase K (10 mg/ml) for 2 h at 55 °C. DNA was purified using phenol:chloroform:isoamyl alcohol extraction, followed by ethanol precipitation. The purified DNA was used for qPCR analysis with locus-specific primers (Additional File 10: Table S9) according to the qPCR protocol described earlier.

### Quantitative mass spectrometry

Treated adherent astrocytes were lysed and subjected to tryptic protein digest using a modified FASP protocol [104]. Proteomic measurements were performed on a LTQ Orbitrap XL mass spectrometer (Thermo Scientific) online coupled to an Ultimate 3000 nano-HPLC (Dionex). Peptides were enriched on a nano trap column (100 µm i.d. $\times$ 2 cm, packed with Acclaim PepMap100 C18, 5 µm, 100 Å, Dionex) prior to separation on an analytical C18 PepMap column (75 µm i.d. $\times$ 25 cm, Acclaim PepMap100 C18, 3 µm, 100 Å, Dionex) in a 135-min linear acetonitrile gradient from 3 to 34% ACN. From the high-resolution orbitrap MS pre-scan (scan range 300–1500 m/z), the ten most intense peptide ions of charge $\geq +2$ were selected for fragment analysis in the linear ion trap if they exceeded an intensity of at least 200 counts. The normalized collision energy

for CID was set to a value of 35. Every ion selected for fragmentation was excluded for 30 s by dynamic exclusion. The individual raw-files were loaded to the Progenesis software (version 4.1, Waters) for label-free quantification and analyzed as described [105, 106]. MS/MS spectra were exported as Mascot generic file and used for peptide identification with Mascot (version 2.4, Matrix Science Inc., Boston, MA, USA) in the Ensembl Mouse protein database (release 75, 51,765 sequences). Search parameters used were as follows: 10 ppm peptide mass tolerance and 0.6 Da fragment mass tolerance, one missed cleavage allowed, carbamidomethylation was set as fixed modification, methionine oxidation and asparagine or glutamine deamidation were allowed as variable modifications. A Mascot-integrated decoy database search was included. Peptide assignments were filtered for an ion score cut-off of 30 and a significance threshold of $p < 0.01$ and were reimported into the Progenesis software. After summing up the abundances of all peptides allocated to each protein, resulting normalized protein abundances were used for calculation of fold-changes and corresponding $p$-values.

### Expression plasmids

In order to overexpress different neurogenic transcription factors in the astroglial cells, we used Moloney murine leukemia virus (MMLV)-derived retroviral vectors, expressing neurogenic fate determinants under the regulatory control of a strong and silencing-resistant pCAG promoter. All our construct encode a neurogenic factor followed by an internal ribosomal entry site (IRES) and either GFP or dsRED as reporter proteins, allowing simultaneous reporter expression. For control experiments, we used a retrovirus encoding for the fluorescent proteins (GFP or dsRED) behind the IRES driven by the same CAG promoter. We used the following expression vectors: pCAG-IRES-GFP [41]; pCAG-IRES-dsRED [41]; pCAG-Neurog2-IRES-dsRED [41]; pCAG-Pou3f2 -IRES-dsRED [107]; pCAG-Sox11-IRES-GFP [44]; pCAG-Hmgb2-IRES-GFP[this work].

### Cloning pCAG-Hmgb2-IRES-GFP construct

cDNA for Hmgb2 were synthetized at Genscript, containing BamHI and HindIII, in order to clone them into the pENTR1A entry vector. The cDNAs were then transferred to the retroviral destination vector pCAG-IRES-dsRED/GFP using the Gateway cloning method (Invitrogen) according to the manufacturer's instructions. The correct sequence was confirmed using Sanger sequencing before viral vector production.

### Retroviral vector production

The VSV-G-pseudotyped retroviruses were prepared using the HEK293-derived retroviral packaging cell line (293GPG) (Ory et al., 1996) that stably express the gag-pol genes of murine leukemia virus and vsv-g under the control of a tet/VP16 transactivator as previously described (Heinrich et al., 2011). The viral particles were stored in TNE (Tris−HCl pH = 7.8 (50 mM); NaCl (130 mM); EDTA (1 mM)) buffer at −80 °C until use.

### Statistical analysis

Numbers of biological replicates can be seen on the dot plots or in the figure legend in case of the bar charts. All results are presented as median ± interquartile range (IQR). IQR was calculated in RStudio [108], using the default method based on type 7

continuous sample quantile. For the reprogramming experiments, statistical analysis was performed in Origin using non-parametric Mann–Whitney *U* test unless differently specified for particular experiments.

## Supplementary Information

Additional File 1: Supplementary Figures. This file contains all supplementary figures

Additional File 2: Table S1. GO analysis of processes enriched in the EGF+bFGF and bFGF only proteomes

Additional File 3: Table S2. Full list of differentially regulated genes between different conditions

Additional File 4: Table S3. GO analysis associated with RNA-seq analysis

Additional File 5: Table S4. Full list of MAS and DAS with their genomic location

Additional File 6: Table S5. GO analysis associated with ATAC analysis

Additional File 7: Table S6. GO analysis associated ATAC peaks enriched in different reprogramming conditions

Additional File 8: Table S7. Full list of MAS and DAS with Neurog2 and Pou TF binding motifs

Additional File 9: Table S8. List of qPCR primers used for assessment of culture purity

Additional File 10: Table S9. List of ATAC-peaks used for ChIP-qPCR validation including qPCR primers

Additional File 11: Peer review history

### Acknowledgements
We thank all members of the Neurogenesis and Regeneration group for experimental input, discussions, and critical reading of the manuscript. We acknowledge the support of the following core facilities: the Bioimaging Core Facility at the BioMedical Center of LMU Munich and the Sequencing Facility at the Helmholtz Zentrum München.

### Peer review information
Sophie Nicod and Tim Sands were the primary editors of this article and managed its editorial process and peer review in collaboration with the rest of the editorial team.

### Authors' contributions
P.M., T.L., and J. N. conceived the project and designed experiments. A.S.-M., V.S., F.B., and J.N. performed experiments. J. M.-P. and S.M.H. analyzed proteome. L.R. and M. B. provided Hmgb2 KO animals. P. M. and J.N. wrote the manuscript with input from all authors.

### Funding
 This work was supported by the German research foundation (DFG) through SFB 870 (J.N. and M.G.); TRR274/1 (ID 408885537) (J.N.); SPP 1738 "Emerging roles of non-coding RNAs in nervous system development, plasticity & disease" (J.N.); SPP1757 "Glial heterogeneity" (J.N.); the Fritz Thyssen Foundation (J.N.); SPP2191 "Molecular mechanisms of functional phase separation" (ID 402723784, project number 419139133) (J.N.); SPP1935 "Deciphering the mRNP code: RNA-bound determinants of post-transcriptional gene regulation" (J.N.); Research Grant Program: Cellular Dynamics and Regulatory Pathways for Successful Regeneration (GO 640/19-1) (J.N. and M.G.); ERC Chrono Neurorepair (M.G.) and the Graduate School for Systemic Neurosciences GSN-LMU (V.S., F.B., P.M. and T.L.).

### Data availability
Proteome data set is available at PRIDE database (https://www.ebi.ac.uk/pride/) [109] with accession number PXD044288. The RNAseq and ATACseq datasets are available at Gene Expression Omnibus (GEO) [110–112] with accession numbers GSE240860, GSE240857, and GSE240782. Images used for the figure are deposited in the FigShare repository [113].

## Declarations

### Ethics approval and consent to participate
All animal experimental procedures were performed in accordance with the German and European Union guidelines and were approved by the Institutional Animal Care and Use Committee (IACUC) and the Government of Upper Bavaria under license number: AZ 55.2–1-54–2532-171–2011 and AZ 55.2–1-54–2532-150–11. All efforts were made to minimize animal suffering and to reduce the number of animals used.

### Competing interests
The authors declare that they have no competing interest.

### Author details
[1]Department of Cell Biology and Anatomy, Biomedical Center Munich (BMC), Medical Faculty, LMU, Munich, Germany. [2]Graduate School of Systemic Neurosciences, LMU, Munich, Germany. [3]Institute of Stem Cell Research, Helmholtz Zentrum Munich, Munich, Germany. [4]Research Unit Central Nervous System Regeneration, Helmholtz Centre Munich,

German Research Center for Environmental Health, Neuherberg, Germany. [5]Research Unit Protein Science and Metabolomics and Proteomics Core, Helmholtz Centre Munich, German Research Center for Environmental Health, , Neuherberg, Germany. [6]School of Medicine, Vita-Salute San Raffaele University, Milan, Italy. [7]Division of Genetics and Cell Biology, IRCCS San Raffaele Hospital, Milan, Italy. [8]Biomedical Center Munich (BMC), Institute of Physiological Genomics, LMU, Munich, Germany. [9]Munich Cluster for Systems Neurology SYNERGY, LMU, Munich, Germany.

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

## 