## [Additional File 11: Peer review history · Genome Biology]

Review history

First round of review

Reviewer 1

In this manuscript, "Hmgb2 improves astrocyte to neuron conversion by increasing the chromatin accessibility of genes associated with neuronal maturation in a proneuronal factor-dependent manner" by Maddhesiya et al., the authors identified the mechanism of how culture conditions affect direct conversion efficiency from astrocytes to neurons in vitro. Their identified molecule, Hmgb2 supports all hypotheses raised by them and explains well what they observed in the different culture conditions. Their experiments were nicely done, and the finding has a great conceptual advance in the relevant field. However, there are some issues that should be addressed prior to publication to make this manuscript further worthwhile and more suitable for publication in this journal.

Specific comments

1. Regarding cells cultured in the presence of EGF and bFGF, is it appropriate to call them astrocytes? They are GFAP-positive but also proliferative. Are they neural progenitors which are converted from astrocytes with EGF+bFGF. If so, their neuronal conversion cannot be referred to as direct reprogramming. Please provide clear differences between astrocytes used in this study and neural progenitors.
2. As for lines 293 to 296, the rationale is not clear as to why the authors thought that post-translational modification is responsible for the suppression of the gene by Neurog2+Hmgb2 overexpression. Please explain it in more detail.
3. Why Neurog2 can alter the chromatin accessibility of neuronal genes only in the presence of Hmgb2? The authors have already denied the possibility that Neurog2 interacts with Hmgb2, but how about: does Hmgb2 alter chromatin modification such as H3K27me3, enabling Neurog2 to bind the target region more efficiently? This reviewer suggests that the author perform experiments to address this issue.
4. Since the authors described the method for "Stab wound injury" in the Material and Methods section, although there were no results shown, this reviewer assumes that the authors are able to examine the function of Hmgb2 in the real injury condition in the brain. This study started based on their interest in the effect of injury-induced neuroinflammatory environment and released growth factors on the direct conversion process, this reviewer suggests, if they are doable, that the author perform some experiments such as, 1) overexpress Hmgb2 in the intact brain and check the astrocyte to neuron conversion efficiency induced by Neurog2 expression, 2) examine Neurog2-induced direct reprogramming efficiency in the brains of Hmgb2 KO or Hmgb2 knock-down mice after stab wound injury.

Minor points

1. There are many typos (e.g. EFG instead of EGF) and mislabeling of Figures (e.g. Figure 5 in the main text is Figure 6). Check and correct them appropriately.

Reviewer 2

The use of neuronal conversion strategies (namely of astrocytes) is a promising approach to tackle pathologies associated with neuronal loss, including traumatic brain injury. This will require a better understanding of the many hurdles facing neuronal reprogramming in the injured brain environment. Astrocyte cultures from postnatal brain have been an important model to study mechanisms of neuronal fate conversion mediated by proneuronal transcription factors. However, standard culture conditions include EGF and bFGF, which may not always reflect the dynamic environment during brain injury. The manuscript by Maddhesiya et al aims at dissecting the impact of growth factors EGF and bFGF on reprogramming mediated by Ngn2. The authors

found Hmgb2 to be dynamically regulated by growth factors, the expression of which, they argue, is required for efficient Ngn2 mediated reprogramming. By combining expression profiling with chromatin accessibility studies, they propose a model whereby Hmgb2 facilitates Ngn2 activity, by increasing accessibility at subset of targets of Ngn2 (and possibly at targets of other downstream TFs). Overall, these are very significant findings for the field of neuronal reprogramming, in particular on how signalling pathways characteristic of the injured brain may impact cell fate conversion. There are however a series of points that should be addressed, for the manuscript to properly support the importance of Hmgb2 in this reprogramming paradigm, and the underlying mechanisms.

The authors show that "EGF only" condition results in the same efficiency of reprogramming as when both mitogens are added, whereas "FGF only" results in almost no reprogramming. My interpretation is that EGF is required for lineage conversion. However, the authors argue instead for an inhibitory effect of bFGF ("...in line with the specific role of bFGF in decreasing the conversion rate" - line 34). Later (line 258) it reads "bFGF culture established the lineage barrier" (here the role of bFGF is ambiguous), but again in Discussion it reads "Our model predicted that prolonged injury-induced elevation in bFGF levels decreased the reprogrammability...". This is a critical point that should be clarified.

When playing with different growth factor conditions, the authors observe that refraction to reprogramming by Ngn2 in astrocytes cultured initially in "bFGF only" conditions, can be rescued by subsequent culture in "EGF+FGF" medium. Does this rescue correlate with increased Hmgb2 expression? Conversely, does shifting astrocytes from "EGF+bFGF" to "bFGF only" conditions result in decreased expression of Hmgb2?

Hmgb2 loss-of-function in astrocytes from Hmgb2 KO cells (MUT cells) are used to show the requirement for this factor. The comparison shown in Figure 3D shows a relatively small (although significant) difference between WT and MUT cells, based on DCX expression. It is important that the authors characterize better existing differences (e.g. neurite complexity). One possibility is that HMGB1 partially compensates for the loss of HMGB2. Is this factor upregulated in MUT cells? This is an important point that should be more thoroughly investigated and discussed.

The expression of Hmgb2 in reactive astrocytes in the context of TBI is very striking (Figure S4). Can the authors in some way provide a more direct link (even if correlative) with growth factor signalling in this context? In cultured astrocytes, Hmgb2 expression was performed at cell population level only. This should be better characterized by immunocytochemistry. Is Hmgb2 homogeneously expressed?

The authors favour a model in which Hmgb2 facilitates "chromatin accessibility and Ngn2 binding" (e.g. line 404), also for TFs downstream Ngn2. This model is speculative at this point, as it is based solely on the enrichment of consensus binding motifs at sites of increased accessibility. This is a crucial point that must be substantiated with mechanistic insights. I suggest two lines of evidence: i) comparing Ngn2 recruitment to selected target sites in astrocytes grown in "bFGF only" and "EGF + bFGF" conditions, and ii) showing evidence of recruitment of Hmgb2 to selected Ngn2 sites (Hmgb2 recruitment has been shown in other publications, for example using chromatin immunoprecipitation).

Typos:

Line 287, Figure 3G should be Figure 4G

Line 486: Is this data not shown?

Figure 1c: Maybe colours are missing in legend (every word is in green)

Authors' response to reviewers

Point by point response to reviewers:

Reviewer #1: In this manuscript, "Hmgb2 improves astrocyte to neuron conversion by increasing the chromatin accessibility of genes associated with neuronal maturation in a proneuronal factor-dependent manner" by Maddhesiya et al., the authors identified the mechanism of how culture conditions affect direct conversion efficiency from astrocytes to neurons in vitro. Their identified molecule, Hmgb2 supports all hypotheses raised by them and explains well what they observed in the different culture conditions. Their experiments were nicely done, and the finding has a great conceptual advance in the relevant field. However, there are some issues that should be addressed prior to publication to make this manuscript further worthwhile and more suitable for publication in this journal.

We sincerely thank the reviewer for the positive assessment and constructive feedback on our manuscript. We believe that we can address the points as outlined below:

Specific comments

1. Regarding cells cultured in the presence of EGF and bFGF, is it appropriate to call them astrocytes? They are GFAP-positive but also proliferative. Are they neural progenitors which are converted from astrocytes with EGF+bFGF. If so, their neuronal conversion cannot be referred to as direct reprogramming. Please provide clear differences between astrocytes used in this study and neural progenitors.

This is an important point. The astrocytes used in the EGF+bFGF astrocyte culture were generated following a widely used protocol (Heinrich et al., 2012; doi: 10.1007/978-1-61779-452-0_32) and have been characterized at the single-cell level in Pereira et al. (2024). Both studies provided detailed analyses showing the absence of neuronal progenitors in these astrocyte cultures. However, the bFGF astrocyte culture has not been characterized previously.

To address this, we performed immunohistochemical analyses of astrocyte and neuronal progenitor markers (Suppl. Fig. 2, previously included in the earlier manuscript). These analyses revealed no differences between the two astrocyte culture conditions. Additionally, we now include new qPCR data comparing neuronal progenitor markers between the two astrocyte cultures and embryonic mouse cerebral cortex. The qPCR results confirm an enrichment of the glial marker *Gfap* in astrocyte cultures (irrespective of growth conditions) compared to the embryonic cortex, along with an underrepresentation of neuronal markers. Importantly, no differences were observed between the two astrocyte culture conditions. These findings strengthen our confidence that there are no neuronal progenitors in our astrocyte cultures. The new data is included in Suppl. Fig. 3b.

2. As for lines 293 to 296, the rationale is not clear as to why the authors thought that post-translational modification is responsible for the suppression of the gene by Neurog2+Hmgb2 overexpression. Please explain it in more detail.

We observed that HMGB2 increases the expression of genes associated with the GO term category linked to phosphorylation (Fig. X). Furthermore, previous studies have demonstrated that Neurogenin's ability to bind its target promoters is influenced by its phosphorylation state (Quan et al., 2016; Ali et al., 2011; Li et al., 2012; Pereira et al., 2024). Based on these findings, we propose that phosphorylation could represent an additional mechanism contributing to the enhanced direct conversion rate observed in the EGF+bFGF culture. This explanation is now explicitly included in the manuscript (lines 308 to 311).

3. Why Neurog2 can alter the chromatin accessibility of neuronal genes only in the presence of Hmgb2? The authors have already denied the possibility that Neurog2 interacts with Hmgb2, but how about: does Hmgb2 alter chromatin modification such as H3K27me3, enabling Neurog2 to bind the target region more efficiently? This reviewer suggests that the author perform experiments to address this issue.

We agree that this is an important aspect. To address it, we conducted experiments analyzing H3K27me3 levels in astrocytes following HMGB2 overexpression in both bFGF and EGF+bFGF cultures. However, we did not observe any differences. Since H3K27me3 represents only one possible modification and does not provide a clear explanation, we believe it is not meaningful to include these results in the manuscript at this stage.

4. Since the authors described the method for "Stab wound injury" in the Material and Methods section, although there were no results shown, this reviewer assumes that the authors are able to examine the function of Hmgb2 in the real injury condition in the brain. This study started based on their interest in the effect of injury-induced neuroinflammatory environment and released growth factors on the direct conversion process, this reviewer suggests, if they are doable, that the author perform some experiments such as, 1) overexpress Hmgb2 in the intact brain and check the astrocyte to neuron conversion efficiency induced by Neurog2 expression, 2) examine Neurog2-induced direct reprogramming efficiency in the brains of Hmgb2 KO or Hmgb2 knock-down mice after stab wound injury.

We performed a stab wound injury and analyzed HMGB2 expression at 5 days post-injury to show that the injury environment temporarily induces HMGB2 expression. This data was included in the original Suppl. Fig. 4. To make this point clearer, we have now extended the analysis to include additional time points (24 hours, 72 hours, and 7 days after injury). Additionally, we infused EGF onto the brain surface of uninjured mice and observed an upregulation of HMGB2, suggesting that EGF induces HMGB2 expression. This new data is now included in the revised manuscript in Suppl. Fig. 6.

Regarding the in vivo direct conversion experiments, we find them to be very interesting and plan to conduct them in the near future. However, we believe that these experiments are beyond the scope of the current manuscript. Additionally, conducting these experiments would require approval for animal studies, which is currently taking up to 9 months to be granted in Munich after submission to the authorities. As such, it would not be feasible to conduct these experiments in time for this manuscript revision.

Minor points

1. There are many typos (e.g. EFG instead of EGF) and mislabeling of Figures (e.g. Figure 5 in the main text is Figure 6). Check and correct them appropriately.

We carefully checked manuscript for typos and correct them.

Reviewer #2: The use of neuronal conversion strategies (namely of astrocytes) is a promising approach to tackle pathologies associated with neuronal loss, including traumatic brain injury. This will require a better understanding of the many hurdles facing neuronal reprogramming in the injured brain environment. Astrocyte cultures from postnatal brain have been an important model to study mechanisms of neuronal fate conversion mediated by proneuronal transcription factors. However, standard culture conditions include EGF and bFGF, which may not always reflect the dynamic environment during brain injury. The manuscript by Maddhesiya et al aims at dissecting the impact of growth factors EGF and bFGF on reprogramming mediated by Ngn2. The authors found Hmgb2 to be dynamically regulated by growth factors, the expression of which, they argue, is required for efficient Ngn2 mediated reprogramming. By combining expression profiling with chromatin accessibility studies, they propose a model whereby Hmgb2 facilitates Ngn2 activity, by increasing accessibility at subset of targets of Ngn2 (and possibly at targets of other downstream TFs). Overall, these are very significant findings for the field of neuronal reprogramming, in particular on how signalling pathways characteristic of the injured brain may impact cell fate conversion. There are however a series of points that should be addressed, for the manuscript to properly support the importance of Hmgb2 in this reprogramming paradigm, and the underlying mechanisms.

1. The authors show that "EGF only" condition results in the same efficiency of reprogramming as when both mitogens are added, whereas "FGF only" results in almost no reprogramming. My interpretation is that EGF is required for lineage conversion. However, the authors argue instead for an inhibitory effect of bFGF ("...in line with the specific role of bFGF in decreasing the conversion rate" - line 34). Later (line 258) it reads "bFGF culture established the lineage barrier" (here the role of bFGF is ambiguous), but again in Discussion it reads "Our model predicted that prolonged injury-induced elevation in bFGF levels decreased the reprogrammability...". This is a critical point that should be clarified.

We agree with our reviewer that "EGF only" is sufficient to generate the culture prone to direct conversion and therefore it is required for the conversion process. We indeed show it in the Suppl. Fig. 1d-f. Moreover, we now included the data that the application of EGF in the intact mouse cerebral cortex induces the expression of the conversion relevant HMGB2. We now changed our statements as follows:

- a) Line 132-135: "Importantly, Neurog2 induced the conversion of astrocytes grown with only EGF at the same rate as astrocytes grown in EGF+bFGF culture medium (Suppl. Fig. 1d-f), in line with the specific role of EGF in decreasing the lineage barrier and promoting direct neuronal conversion."
- b) Line 274-278: "Therefore, EGF-induced HMGB2 likely lowers the conversion barrier by regulating the expression of a small, specific set of genes that are crucial for the conversion process. To test this hypothesis, we selected one candidate gene, Prox1, and investigated whether its expression could help overcome the lineage barrier in the bFGF-only medium, which is characterized by low HMGB2 levels."
- c) Line 469-472: "Our model suggests that exposure to EGF is essential for conversion in the injury-induced environment. In contrast, the prolonged elevation of bFGF levels may contribute to the low conversion rate, as FGF signaling alone promotes processes related to neurogenesis and neuronal fate in astrocytes during Neurog2-mediated conversion."

2. When playing with different growth factor conditions, the authors observe that refraction to reprogramming by Ngn2 in astrocytes cultured initially in "bFGF only" conditions, can be rescued by subsequent culture in "EGF+FGF" medium. Does this rescue correlate with increased Hmgb2 expression? Conversely, does shifting astrocytes from "EGF+bFGF" to "bFGF only" conditions result in decreased expression of Hmgb2?

These are indeed important experiments that would further validate our model, and as such, we conducted them. Consistent with our reprogramming results, the levels of HMGB2 varied according to the reprogrammability of the culture. These data are now included in Suppl. Figure 5.

3. Hmgb2 loss-of-function in astrocytes from Hmgb2 KO cells (MUT cells) are used to show the requirement for this factor. The comparison shown in Figure 3D shows a relatively small (although significant) difference between WT and MUT cells, based on DCX expression. It is important that the authors characterize better existing differences (e.g. neurite complexity). One possibility is that HMGB1 partially compensates for the loss of HMGB2. Is this factor upregulated in MUT cells? This is an important point that should be more thoroughly investigated and discussed.

We analyzed HMGB1 expression in HMGB2 knockout (KO) animals and found no increase, suggesting that HMGB1 is unlikely to compensate for the loss of HMGB2 in these animals. These data are now included in Fig. 3e-i. Additionally, we conducted a pilot experiment to investigate whether HMGB1 could improve reprogramming in the bFGF culture, but we found no effect. These results are provided in a figure for the reviewer. Furthermore, we analyzed neurite morphology in HMGB2-deficient animals and observed reduced complexity compared to wild-type siblings. The Sholl analysis showing these results is now included in Fig. 3j, k.

4. The expression of Hmgb2 in reactive astrocytes in the context of TBI is very striking (Figure S4). Can the authors in some way provide a more direct link (even if correlative) with growth factor signalling in this context? In cultured astrocytes, Hmgb2 expression was performed at cell population level only. This should be better characterized by immunocytochemistry. Is Hmgb2 homogeneously expressed?

We analyzed HMGB2 expression in cultured astrocytes using immunocytochemistry to assess the heterogeneity of HMGB2 expression in our culture system. These data are shown in Suppl. Fig. 3c-g. While we observed some variability in expression, HMGB2 levels were generally lower in the bFGF culture, consistent with findings from Western blot (WB) and mass spectrometry.

The dynamics of growth factor expression following stab wound injury have been described by Addington et al., who reported that EGF levels peak at 24 hours, while bFGF remains elevated for up to 14 days. We analyzed HMGB2 expression at 1, 3, and 7 days post-injury and observed a peak in expression between 3 and 5 days, which aligns with the possibility that EGF induces HMGB2 expression. Furthermore, we applied EGF directly to the brain surface of uninjured mice and observed the induction of HMGB2, along with mild astrogliosis. These data are included in Suppl. Fig. 6.

5. The authors favour a model in which Hmgb2 facilitates "chromatin accessibility and Ngn2 binding" (e.g. line 404), also for TFs downstream Ngn2. This model is speculative at this point, as it is based solely on the enrichment of consensus binding motifs at sites of increased accessibility. This is a crucial point that must be substantiated with mechanistic insights. I suggest two lines of evidence: i) comparing Ngn2 recruitment to selected target sites in astrocytes grown in "bFGF only" and "EGF + bFGF" conditions, and ii) showing evidence of recruitment of Hmgb2 to selected Ngn2 sites (Hmgb2 recruitment has been shown in other publications, for example using chromatin immunoprecipitation).

We performed the ChIP-qPCR experiments to address if both Neurog2 and Hmgb2 are binding these sites. The data is now included in the Fig. 6f, g.

Typos:

Line 287, Figure 3G should be Figure 4G

We corrected it in the revised manuscript.

Line 486: Is this data not shown?

This data is not shown, and that is correct. We believe that including it in the manuscript may lengthen it without providing additional explanatory value. However, we are happy to include it if the reviewer and editors feel it is necessary.

Figure 1c: Maybe colours are missing in legend (every word is in green)

We corrected this mistake in the revised manuscript.

Second round of review

Reviewer 2

After reading the revised version of the manuscript, I consider that the authors have addressed extensively, either experimentally or by writing, all the critical points that I have previously raised. Also, there is no need to include the data from experiments probing a direct interaction between Hmgb2 and Ngn2 ("data not shown" is enough).